# THE SPARSE MATRIX-BASED RANDOM PROJECTION: EXPLORING OPTIMAL SPARSITY FOR CLASSIFICATION

## ABSTRACT

In the paper, we study the sparse $\{0, \pm 1\}$-matrix-based random projection, a technique extensively applied in diverse classification tasks for dimensionality reduction and as a foundational model for each layer in the popular deep ternary networks. For these sparse matrices, determining the optimal sparsity level, namely the minimum number of nonzero entries $\pm 1$ needed to achieve the optimal or near-optimal classification performance, remains an unresolved challenge. To investigate the impact of matrix sparsity on classification, we propose to analyze the mean absolute deviation (MAD) of projected data points, which quantifies their dispersion. Statistically, a higher degree of dispersion is expected to improve classification performance by capturing more intrinsic variations in the original data. Given that the MAD value depends not only on the sparsity level of random matrices but also on the distribution of the original data, we evaluate two representative data distributions for generality: the Gaussian mixture distribution, widely used to model complex real-world data; and the two-point distribution, available for modeling discretized data. Our analysis reveals that sparse matrices with only *one* or *a few* nonzero entries per row can achieve MAD values comparable to, or even exceed, those of denser matrices, provided the matrix size satisfies $m \geq \mathcal{O}(\sqrt{n})$, where $m$ and $n$ denote the projected and original dimensions, respectively. These extremely sparse matrix structures imply significant computational savings. This finding is further validated through classification experiments on diverse real-world datasets, including images, text, gene data, and binary-quantized data, demonstrating its broad applicability.

## 1 INTRODUCTION

Random projection is a fundamental, unsupervised dimensionality reduction technique that projects high-dimensional data onto low-dimensional subspaces using random matrices (Johnson & Lindenstrauss, 1984). This projection method can approximately maintain the pairwise $\ell_2$ distance between original data points, preserving the structure of original data and making it suitable for classification tasks (Bingham & Mannila, 2001; Fradkin & Madigan, 2003; Wright et al., 2009). To ensure the preservation of $\ell_2$ distances, random projection matrices must adhere to specific distributions, such as Gaussian matrices (Dasgupta & Gupta, 1999) and sparse $\{0, \pm 1\}$-ternary matrices (hereinafter referred to as sparse matrices) (Achlioptas, 2003). In practical scenarios, sparse matrices are favored due to their significantly lower storage and computational complexity. Given that random projection is extensively used in computationally intensive, large-scale classification tasks (Gionis et al., 1999), and can even model each layer of deep networks (Giryes et al., 2016), it is highly desirable to investigate the optimal sparsity of sparse matrices, namely the minimum number of nonzero entries $\pm 1$ required for the projected data to achieve the optimal or near-optimal classification performance. To the best of our knowledge, this problem has not been previously investigated.

Existing random projection research has primarily focused on characterizing the distribution of random matrices that effectively preserve pairwise distances. Specifically, these matrices are designed to maintain the expected pairwise distances between original data points after projection, while keeping the variance relatively small (Dasgupta & Gupta, 1999; Achlioptas, 2003). For sparse matrices with appropriately scaled entries, distance preservation has been validated in the $\ell_2$ norm (Achlioptas, 2003; Li et al., 2006), but fails in the $\ell_1$ norm (Brinkman & Charikar, 2003; Li, 2007). While the $\ell_2$ distance preservation property supports classification tasks, it does not lend itself well

to analyzing the influence of matrix sparsity (i.e., the number of nonzero entries) on subsequent classification accuracy. This is because classification accuracy depends on the discriminative power among projected points rather than strict preservation of original data structure. For example, $\ell_2$ distance preservation degrades as the matrix becomes sparser, namely containing fewer nonzero entries $\pm 1$ (Li et al., 2006). However, empirical studies reveal that sparser matrices does not necessarily lead to worse classification performance; in fact, extremely sparse matrices, such as those with only one or a few dozen nonzero entries per row, often achieve comparable or superior classification accuracy compared to denser alternatives. This counterintuitive phenomenon remains theoretically unexplained.

In the paper, we demonstrate that the performance advantages of extremely sparse matrices can be explained by analyzing the dispersion of projected data points. Statistically, a higher degree of dispersion should lead to improved classification performance, as it captures more intrinsic variations in the original data (Jolliffe & Cadima, 2016). Generally, dispersion is quantified using variances, as seen in principal component analysis (PCA) (Jolliffe, 2002), primarily due to the computational and analytical convenience of variances. However, variance-based dispersion analysis is ideally suited for Gaussian data and sensitive to noise and outliers (Hubert et al., 2016). Such limitations are common in real-world applications. In these cases, the mean absolute deviation (MAD) (Yager & Alajlan, 2014) offers a more robust alternative for quantifying dispersion (Deng et al., 2007; McCoy & Tropp, 2011). Consequently, we adopt MAD in our study to identify the matrix sparsity that maximizes dispersion, thereby enhancing classification performance.

The MAD value depends not only on the sparsity level of random matrices, but also on the distribution of the original data. For the sake of generality, we consider two representative data distributions: the Gaussian mixture distribution, which has been extensively used to model natural data distributions (Torralba & Oliva, 2003; Weiss & Freeman, 2007) and their feature transformations (Wainwright & Simoncelli, 1999; Lam & Goodman, 2000); and the two-point distribution, which is suitable for modelling discretized data, a scenario gaining increasing relevance with the emergence of large-scale data and models (Gionis et al., 1999; Hubara et al., 2016; Yang et al., 2019). Benefiting from the two fundamental distributions, as demonstrated later, our theoretical analysis results can be generalized to a broad range of real-world data scenarios.

By analyzing the MAD of projected data points across varying levels of matrix sparsity, we identify two major results. First, sparse matrices with exactly one nonzero entry per row tends to achieve best classification performance, as the original data exhibit sufficiently high discrimination between individuals. Second, as matrix sparsity increases, classification accuracy converges asymptotically to a stable plateau; notably, this convergence occurs even with moderate sparsity levels, such as the level of containing only a few dozen nonzero entries per row. Both findings hold under the condition $m \geq \mathcal{O}(\sqrt{n})$, where $m$ and $n$ denote the projected and original dimensions, respectively. Collectively, the above two results demonstrate that extremely sparse matrices with only *one* or *a few* nonzero entries per row perform comparably or superiorly to denser alternatives in classification tasks. This performance advantage is empirically validated across various real-world datasets, including images, texts, gene expression data, and binary-quantized data. Crucially, our findings imply that the computational complexity of sparse matrices-based random projection can be drastically reduced, while maintaining or even enhancing downstream classification accuracy. This breakthrough represents a significant efficiency improvement for random projection-based models, particularly for high-dimensional data applications.

## 2 PROBLEM FORMULATION

Consider a random projection $z = Rh$, where $h \in \mathbb{R}^n$ denotes an original data and $R \in \{0, \pm 1\}^{m \times n}$ is a sparse random matrix. In the paper, we aim to estimate the optimal sparsity level of $R$, namely the minimum number of nonzero entries $\pm 1$, that enables maximizing the mean absolute deviation (MAD) of projected data points $z \in \mathbb{R}^m$, formally defined as $\text{MAD}(z) = \mathbb{E}\|z - \mathbb{E}z\|_1$. As discussed before, a larger $\text{MAD}(z)$ is expected to yield better classification performance. Since $\text{MAD}(z)$ involves both the distributions of random matrices $R$ and original data $h$, in the sequence we first specify their probabilistic models before detailing the MAD estimation procedure.

**Notation.** Throughout the paper, we typically denote a matrix by a bold upper-case letter $R \in \mathbb{R}^{m \times n}$, a vector by a bold lower-case letter $r = (r_1, r_2, ..., r_n)^\top \in \mathbb{R}^n$, and a scalar by a lower-case

letter $r_i$ or $r$. Sometimes, we use the bold letter $\boldsymbol{r}_i \in \mathbb{R}^n$ to denote the $i$-th row of $\boldsymbol{R} \in \mathbb{R}^{m \times n}$. For ease of presentation, we defer all proofs to the appendix.

## 2.1 THE DISTRIBUTION OF SPARSE MATRICES

The sparse random matrix $\boldsymbol{R}$ we aim to study is specified in Definition 1, which has the parameter $k$ counting the number of nonzero entries per row, and is simply called $k$-sparse to distinguish between the matrices of different sparsity. Instead of the form $\boldsymbol{R} \in \{0, \pm 1\}^{m \times n}$, in the definition we introduce a scaling parameter $\sqrt{\frac{n}{mk}}$ to make the matrix entries have zero mean and unit variance. With this distribution, the matrix will hold the $\ell_2$ distance preservation property, that is, keeping the expected $\ell_2$ distance between original data points unchanged after random projection (Achlioptas, 2003). Note that the scaling parameter can be omitted in practical applications for easier computation; and the omitting will not change the relative distances between projected data points, thus not affecting downstream classification performance.

**Definition 1** ($k$-sparse random matrix). A $k$-sparse random matrix $\boldsymbol{R} \in \{0, \pm\sqrt{\frac{n}{mk}}\}^{m \times n}$ is defined to be of the following structure properties:

- Each row $\boldsymbol{r} \in \{0, \pm\sqrt{\frac{n}{mk}}\}^n$ contains exactly $k$ nonzero entries, $1 \le k \le n$;

- The positions of $k$ nonzero entries are arranged uniformly at random;

- Each nonzero entry takes the bipolar values $\pm\sqrt{\frac{n}{mk}}$ with equal probability.

## 2.2 THE DISTRIBUTION OF ORIGINAL DATA

For the original data $\boldsymbol{h} = (h_1, h_2, \ldots, h_n)^\top$, as discussed previously, we investigate two representative distributions: two-point distributions and Gaussian mixture distributions, which are detailed below. When analyzing $\text{MAD}(\boldsymbol{z})$, the difference $\boldsymbol{x}$ between two arbitrary data points $\boldsymbol{h}$ and $\boldsymbol{h}'$, namely $\boldsymbol{x} = \boldsymbol{h} - \boldsymbol{h}' = (x_1, x_2, \ldots, x_n)^\top$, will be involved. Then the distribution of the difference vector $\boldsymbol{x}$ is also introduced.

### 2.2.1 TWO-POINT MIXTURE DISTRIBUTION

The two-point distribution is modeled as $\boldsymbol{h} \in \{\mu_1, \mu_2\}^n$, $\mu_1 \ne \mu_2$, with each entry $h_i \in \{\mu_1, \mu_2\}$ independently and identically distributed. As mentioned before, this distribution widely exists in quantization tasks, such as the quantization of deep networks (Hubara et al., 2016).

For the difference vector $\boldsymbol{x}$ between two data points $\boldsymbol{h}$ and $\boldsymbol{h}'$, its each entry $x_i$ will follow a ternary discrete distribution

$$x_i \sim \mathcal{T}(\mu, p, q) \tag{1}$$

with the probability mass function $t \in \{-\mu, 0, \mu\}$ under the probabilities $\{q, p, q\}$, where $\mu = \mu_1 - \mu_2$ and $p + 2q = 1$. Note that a smaller $p$ value (equivalently, a larger $q$ value) indicates a greater number of nonzero entries in $x_i$, suggesting a higher degree of discrimination between $h_i$ and $h_i'$.

### 2.2.2 GAUSSIAN MIXTURE DISTRIBUTION

For the Gaussian mixture distribution, we assume two Gaussian components for each entry of the original data $\boldsymbol{h}$: $h_i \sim \Sigma_{j=1}^2 \omega_j \mathcal{N}(\mu_j, \sigma^2)$, where $\mu_1 \ne \mu_2$, $\omega_1 + \omega_2 = 1$, and $\omega_j \ge 0$. With the two components, we can model the distribution of feature values in each dimension as having two underlying states: "strong" and "weak". Each Gaussian component would represent one of these states, with its mean and variance characterizing the typical values and spread for that intensity level. This modeling method is logically sound and applicable to various areas (Rowe & Blake, 1995; Wang et al., 2014; Xing et al., 2001; Mulè et al., 2022; Zapevalov & Knyazkov, 2023), striking a balance between model complexity and generalization ability.

The difference vector $\boldsymbol{x}$ between two data points $\boldsymbol{h}$ and $\boldsymbol{h}'$ has each entry $x_i$ satisfying a three-component Gaussian mixture distribution

$$x_i \sim \mathcal{M}(\mu, \sigma^2, p, q) \tag{2}$$

with the probability density function

$$f(t) = pf_{\mathcal{N}}(t; 0, \sigma^2) + qf_{\mathcal{N}}(t; \mu, \sigma^2) + qf_{\mathcal{N}}(t; -\mu, \sigma^2) \tag{3}$$

where $f_{\mathcal{N}}(t; \mu, \sigma^2)$ denotes the density function of $t \sim \mathcal{N}(\mu, \sigma^2)$, and $p$ and $q$ represent the mixture weights of three components, with $p + 2q = 1$, and $p, q \geq 0$. As in the two-pint distribution equation 1, a smaller $p$ value indicates a higher degree of discrimination between $h_i$ and $h_i'$.

### 2.3 THE MAD ESTIMATION FOR PROJECTED DATA

As noted at the beginning of this section, our objective is to estimate the minimum sparsity $k$ of the matrix $\boldsymbol{R} \in \{0, \pm\sqrt{\frac{n}{mk}}\}^{m \times n}$ that maximizes the MAD($\boldsymbol{z}$) of the projected data $\boldsymbol{z} = \boldsymbol{Rh}$. We now show that analyzing how MAD($\boldsymbol{z}$) varies with $k$ can be simplified to studying $\mathbb{E}|\boldsymbol{r}^\top \boldsymbol{x}|$, where $\boldsymbol{x} = \boldsymbol{h} - \boldsymbol{h}'$ and $\boldsymbol{r}$ is a row of $\boldsymbol{R}$.

By Property 1, when the original data $\boldsymbol{h}$ follow the Gaussian mixture distribution described above, the projected data $\boldsymbol{z}$ remain Gaussian. Moreover, MAD($\boldsymbol{z}$) is a positive constant multiple of $\mathbb{E}|\boldsymbol{z} - \boldsymbol{z}'|_1$, where $\mathbb{E}|\boldsymbol{z} - \boldsymbol{z}'|_1 = \mathbb{E}|\boldsymbol{Rx}|_1$ and $\boldsymbol{x} = \boldsymbol{h} - \boldsymbol{h}'$. This implies that $\mathbb{E}|\boldsymbol{Rx}|_1$ varies in the same way as MAD($\boldsymbol{z}$) when the matrix sparsity $k$ changes. In particular, a larger value of $\mathbb{E}|\boldsymbol{Rx}|_1$ corresponds to a larger MAD($\boldsymbol{z}$). This relationship also holds approximately for original data $\boldsymbol{h}$ drawn from other distributions, since by the Central Limit Theorem, the projected data $\boldsymbol{z} \in \mathbb{R}^m$ can be approximated by a Gaussian distribution. Therefore, rather than analyzing MAD($\boldsymbol{z}$) directly, we can examine how $\mathbb{E}|\boldsymbol{Rx}|_1$ changes with $k$.

Note that the analysis of $\mathbb{E}\|\boldsymbol{Rx}\|_1$ can be further simplified by focusing on $\mathbb{E}|\boldsymbol{r}^\top \boldsymbol{x}|$, where $\boldsymbol{r} \in \mathbb{R}^n$ is a row of $\boldsymbol{R}$, as the latter exhibits the same trend as the former, with respect to variations in $k$. This equivalence arises from the fact that $\mathbb{E}\|\boldsymbol{Rx}\|_1 = m\mathbb{E}|\boldsymbol{r}^\top \boldsymbol{x}|$, since each row $\boldsymbol{r}$ of $\boldsymbol{R}$ is independently and identically distributed by Definition 1. Therefore, $\mathbb{E}|\boldsymbol{r}^\top \boldsymbol{x}|$ is adopted as our metric to characterize the relationship between MAD($\boldsymbol{z}$) and matrix sparsity $k$. A larger $\mathbb{E}|\boldsymbol{r}^\top \boldsymbol{x}|$ value corresponds to a higher MAD($\boldsymbol{z}$), then yielding improved classification performance.

**Property 1.** Consider $\boldsymbol{z} = \boldsymbol{Rh}$, where $\boldsymbol{h} \in \mathbb{R}^n$ follows the Gaussian mixture distribution described in Section 2.2.2 and $\boldsymbol{R} \in \{0, \pm 1\}^{m \times n}$ is distributed as specified in Definition 1, with its scale parameter omitted for brevity. Considering Gaussian distributions are closed under linear transformations, the i.i.d. entries $z_i$ of $\boldsymbol{z}$ still holds a Gaussian distribution: $z_i \sim \mathcal{N}(\mu, \sigma^2)$. For this distribution of $\boldsymbol{z}$, we have $\mathbb{E}\|\boldsymbol{z} - \mathbb{E}\boldsymbol{z}\|_1 = \frac{1}{\sqrt{2}}\mathbb{E}\|\boldsymbol{z} - \boldsymbol{z}'\|_1$, where $\boldsymbol{z}'$ is an independent copy of $\boldsymbol{z}$.

## 3 THEORETICAL RESULTS

### 3.1 TWO-POINT DATA

As analyzed in Section 2.3, the trend of MAD($\boldsymbol{z}$) against varying matrix sparsity $k$ can be estimated with $\mathbb{E}|\boldsymbol{r}^\top \boldsymbol{x}|$. A larger $\mathbb{E}|\boldsymbol{r}^\top \boldsymbol{x}|$ corresponds to a higher MAD($\boldsymbol{z}$), thereby achieving improved classification performance. Based on this relationship, we analyze how $\mathbb{E}|\boldsymbol{r}^\top \boldsymbol{x}|$ changes with respect to $k$, assuming that the original data vectors $\boldsymbol{h}$ and $\boldsymbol{h}'$ are drawn from a two-point distribution, such that their difference vector $\boldsymbol{x} = \boldsymbol{h} - \boldsymbol{h}'$ has i.i.d. entries $x_i \sim \mathcal{T}(\mu, p, q)$ as defined in equation 1.

**Theorem 1.** Let $\boldsymbol{r}$ be a row of a $k$-sparse random matrix $\boldsymbol{R} \in \{0, \pm\sqrt{\frac{n}{mk}}\}^{m \times n}$, and $\boldsymbol{x} \in \mathbb{R}^n$ with i.i.d. entries $x_i \sim \mathcal{T}(\mu, p, q)$. It can be derived that

$$\mathbb{E}|\boldsymbol{r}^\top \boldsymbol{x}| = 2\mu\sqrt{\frac{n}{mk}} \sum_{i=0}^{k} C_k^i p^i q^{k-i} \left\lceil \frac{k-i}{2} \right\rceil C_{k-i}^{\lceil \frac{k-i}{2} \rceil} \tag{4}$$

and

$$\mathrm{Var}(|\boldsymbol{r}^\top \boldsymbol{x}|) = \frac{2q\mu^2 n}{m} - \frac{4\mu^2 n}{mk} \left( \sum_{i=0}^{k} C_k^i p^i q^{k-i} \left\lceil \frac{k-i}{2} \right\rceil C_{k-i}^{\lceil \frac{k-i}{2} \rceil} \right)^2 \tag{5}$$

where $C_k^i$ is a binomial coefficient $\binom{k}{i}$ and $\lceil \alpha \rceil = \min\{\beta : \beta \geq \alpha, \beta \in \mathbb{Z}\}$. By equation 4, $\mathbb{E}|\boldsymbol{r}^\top \boldsymbol{x}|$ satisfies the following two properties:

(P1) When $p \leq 0.188$, $\mathbb{E}|\boldsymbol{r}^\top \boldsymbol{x}|$ has its maximum at $k = 1$.

(P2) When $k \to \infty$, $\mathbb{E}|\boldsymbol{r}^\top \boldsymbol{x}|$ converges to a constant:

$$\lim_{k \to \infty} \frac{\sqrt{m}}{\mu\sqrt{n}} \mathbb{E}|\boldsymbol{r}^\top \boldsymbol{x}| = 2\sqrt{q/\pi}, \tag{6}$$

which has the convergence error upper-bounded by

$$\left| \frac{\sqrt{m}}{\mu\sqrt{n}} \mathbb{E}|\boldsymbol{r}^\top \boldsymbol{x}| - 2\sqrt{q/\pi} \right| \leq \frac{\sqrt{\pi} + \sqrt{2}}{\sqrt{\pi k}}, \tag{7}$$

for finite $k$ values.

**Remarks of Theorem 1:** In P1 and P2, we demonstrate two distinct trends of $\mathbb{E}|\boldsymbol{r}^\top \boldsymbol{x}|$ against varying matrix sparsity $k$, indicating two corresponding changes in classification performance, which are elaborated as follows.

- P1 indicates that the best classification performance can be achieved by very sparse random matrices with sparsity $k = 1$, when the discrimination between data points is sufficiently high. This can be explained as follows. By P1, $\mathbb{E}|\boldsymbol{r}^\top \boldsymbol{x}|$ will achieve its maximum value at $k = 1$, if the probability $p$ of $x_i = 0$ is sufficiently small, namely $p \leq 0.188$. As mentioned in Section 2.2.1, this condition indicates that the difference $\boldsymbol{x}$ between two data points $\boldsymbol{h}$ and $\boldsymbol{h}'$ should contain a sufficient number of nonzero entries, suggesting that the two data points $\boldsymbol{h}$ and $\boldsymbol{h}'$ should be sufficiently distinct from each other. Then we can say that given the data distribution that exhibits sufficiently high discrimination between samples, the best classification performance can be attained using very sparse random matrices with sparsity $k = 1$, in terms of the maximum $\mathbb{E}|\boldsymbol{r}^\top \boldsymbol{x}|$ achieved at $k = 1$.

- P2 implies that the classification performance will become comparable, as the matrix sparsity $k$ increases. This is because as shown in equation 6, $\mathbb{E}|\boldsymbol{r}^\top \boldsymbol{x}|$ will converge to a constant that merely depends on the data distribution and matrix size, as $k$ tends to infinity. Furthermore, the convergence can be achieved when $k$ is small. This is analyzed below. Note that in equation 6 we describe the convergence with $\mathbb{E}|\boldsymbol{r}^\top \boldsymbol{x}|/(\mu\sqrt{n/m})$ instead of $\mathbb{E}|\boldsymbol{r}^\top \boldsymbol{x}|$, in terms of the fact that both formulas share the same changing trend against varying $k$, but the former has fewer parameters, only involving $k$ and $p$. The convergence error, namely the difference between the values of $\mathbb{E}|\boldsymbol{r}^\top \boldsymbol{x}|$ with finite $k$ and infinite $k$, is upper-bounded in equation 7, and the bound indicates a convergence speed $\mathcal{O}(1/\sqrt{k})$. By the bound equation 7, it is easy to further derive that

$$\frac{\left| \frac{\sqrt{m}}{\mu\sqrt{n}} \mathbb{E}|\boldsymbol{r}^\top \boldsymbol{x}| - 2\sqrt{q/\pi} \right|}{2\sqrt{q/\pi}} \leq \eta, \text{ if } k \geq \frac{(\sqrt{\pi} + \sqrt{2})^2}{4q\eta^2} \tag{8}$$

where $\eta$ can be an arbitrary positive constant. It is seen that $\eta$ sets an upper bound for the ratio between the convergence error with the convergence value (hereinafter referred to as the convergence ratio error). For any arbitrarily small upper bound $\eta$, as shown in equation 8, there exists a corresponding minimum sparsity $k$ required to maintain this bound. When $k$ falls within this specified range, $\mathbb{E}|r^\top x|$ assumes similar values, suggesting that these $k$ values should result in similar classification performance. Consequently, sparse matrices with small $k$ values (around the minimum threshold) exhibit classification performance that is on par with denser matrices possessing larger $k$ values. Note that the lower bound of $k$ theoretically derived in equation 8 contains slack, and its actual value should be relatively small, typically on the order of tens, as evidenced by our numerical computations presented in Appendix C.1.

To verify the accuracy of our theoretical results P1 and P2, in Appendix C.1 we further investigate the trend of $\mathbb{E}|\mathbf{r}^\top \mathbf{x}|/(\mu\sqrt{n/m})$ in two ways: 1) by directly computing it with equation 9, and 2) by statistically estimating its value using synthetic data. Both approaches yield results consistent with P1 and P2. Statistical simulations also show that $\text{MAD}(\boldsymbol{z})$ and $\mathbb{E}|\boldsymbol{r}^\top \boldsymbol{x}|$ share similar trends when varying matrix sparsity $k$, validating the equivalence relation between them as demonstrated in Section 2.3.

Finally, it is noteworthy that the theoretical results discussed above are derived based on the *expected* distance $\mathbb{E}\|\boldsymbol{R}\boldsymbol{x}\|_1$ (or equivalently, $m\mathbb{E}|\boldsymbol{r}^\top\boldsymbol{x}|$), rather than the actual distance $\|\boldsymbol{R}\boldsymbol{x}\|_1$ that would be obtained with a single matrix. To approximate the expected distance and consequently attain the theoretical performance, a single matrix should have the size $m \geq \mathcal{O}(\sqrt{n})$ as shown in Property 2.

**Property 2.** Let $\boldsymbol{r}_i$ be the $i$-th row of a $k$-sparse random matrix $\boldsymbol{R} \in \{0, \pm\sqrt{\frac{n}{mk}}\}^{m\times n}$, and $\boldsymbol{x} \in \mathbb{R}^n$ with i.i.d. entries $x_i \sim \mathcal{T}(\mu, p, q)$. Suppose $z = \frac{1}{m}\|\boldsymbol{R}\boldsymbol{x}\|_1 = \frac{1}{m}\sum_{i=1}^m |\boldsymbol{r}_i^\top\boldsymbol{x}|$. For arbitrarily small $\varepsilon, \delta > 0$, we have the probability $\Pr\{|z - \mathbb{E}z| \leq \varepsilon\} \geq 1 - \delta$, if $\frac{m^2}{m+1} \geq \frac{q\mu^2 n}{\varepsilon^2\delta}$; and the condition can be relaxed to $m^2 \geq \frac{2q\mu^2 n}{\varepsilon^2\delta}$, for a given $\boldsymbol{x}$.

## 3.2 Gaussian Mixture Data

Similarly to the analysis in the previous section, by examining $\mathbb{E}|\boldsymbol{r}^\top\boldsymbol{x}|$ against varying matrix sparsity $k$, we investigate the relationship between $\mathrm{MAD}(\boldsymbol{z})$ and $k$. Here, the original data vectors $\boldsymbol{h}$ and $\boldsymbol{h}'$ are drawn from Gaussian mixture distributions, such that their difference $\boldsymbol{x} = \boldsymbol{h} - \boldsymbol{h}'$ has i.i.d. entries $x_i \sim \mathcal{M}(\mu, \sigma^2, p, q)$ as specified in equation 2.

**Theorem 2.** Let $\boldsymbol{r}$ be a row of a $k$-sparse random matrix $\boldsymbol{R} \in \{0, \pm\sqrt{\frac{n}{mk}}\}^{m\times n}$, and $\boldsymbol{x} \in \mathbb{R}^n$ with i.i.d. entries $x_i \sim \mathcal{M}(\mu, \sigma^2, p, q)$. It can be derived that

$$\mathbb{E}|\boldsymbol{r}^\top\boldsymbol{x}| = 2\mu\sqrt{\frac{n}{mk}}T_1 + \sigma\sqrt{\frac{2n}{\pi m}}T_2 - 2\mu\sqrt{\frac{n}{mk}}T_3 \tag{9}$$

$$T_1 = \sum_{i=0}^k C_k^i p^i q^{k-i} \left\lceil \frac{k-i}{2} \right\rceil C_{k-i}^{\lceil\frac{k-i}{2}\rceil}$$

$$T_2 = \sum_{i=0}^k C_k^i p^i q^{k-i} \sum_{j=0}^{k-i} C_{k-i}^j e^{-\frac{(k-i-2j)^2\mu^2}{2k\sigma^2}}$$

$$T_3 = \sum_{i=0}^k C_k^i p^i q^{k-i} \sum_{j=0}^{k-i} C_{k-i}^j \Phi\left(-\frac{|k-i-2j|\mu}{\sqrt{k}\sigma}\right)$$

and

$$\mathrm{Var}(|\boldsymbol{r}^\top\boldsymbol{x}|) = \frac{n}{m}(\sigma^2 + 2q\mu^2) - \left(\mathbb{E}|\boldsymbol{r}^\top\boldsymbol{x}|_1\right)^2 \tag{10}$$

where $\Phi(\cdot)$ is the distribution function of $\mathcal{N}(0,1)$. Further, we have

$$\mathbb{E}|\boldsymbol{r}^\top\boldsymbol{x}| \leq \mu\sqrt{\frac{n}{m}} + \sigma\sqrt{\frac{2n}{\pi m}}, \tag{11}$$

and

$$\lim_{k\to\infty}\frac{\sqrt{m}}{\mu\sqrt{n}}\mathbb{E}|\boldsymbol{r}^\top\boldsymbol{x}| = \sqrt{\frac{2}{\pi}(\sigma^2 + 2q\mu^2)} \tag{12}$$

which has the convergence error for finite $k$ upper-bounded by

$$\left|\frac{\sqrt{m}}{\mu\sqrt{n}}\mathbb{E}|\boldsymbol{r}^\top\boldsymbol{x}| - \sqrt{2(\sigma^2 + 2q\mu^2)/\pi}\right| \leq \frac{4\sigma^3\left[p + 2q(1 + \mu^2/\sigma^2)^{3/2}\right]}{(\sigma^2 + 2q\mu^2)\sqrt{\pi k}} + \frac{\sqrt{2}[3\sigma^4 + 2q(6\sigma^2\mu^2 + \mu^4)]}{\sqrt{(\sigma^2 + 2q\mu^2)\pi k}}. \tag{13}$$

**Remarks of Theorem 2:** From the results, it can be seen that $\mathbb{E}|\boldsymbol{r}^\top\boldsymbol{x}|$ exhibit similar trends as derived in P1 and P2 of Theorem 1. Specifically:

- Similarly to P1, by numerically computing equation 9 as elaborated in Appendix C.2, it can be observed that $\mathbb{E}|\boldsymbol{r}^\top\boldsymbol{x}|$ will reach its maximum value at $k = 1$, when the data distribution parameter $p$ (specified in equation 2) assumes relatively small values. This suggests that sparse random matrices can achieve the best classification performance at the sparsity level of $k = 1$, when the two data points sampled from the Gaussian-mixture distribution exhibit sufficient discriminability (corresponding to small $p$ values).

- Similarly to P2, by equation 12 and equation 13, it can be deduced that $\mathbb{E}|\boldsymbol{r}^{\top}\boldsymbol{x}|$ will converge to a constant as matrix sparsity $k$ increases, with a convergence rate of $\mathcal{O}(1/\sqrt{k})$. With equation 13, the lower bound of $k$ ensuring the convergence error ratio less than a given constant $\eta$ can be derived:

$$\frac{\left| \frac{\sqrt{m}}{\mu\sqrt{n}} \mathbb{E}|\boldsymbol{r}^{\top}\boldsymbol{x}| - \sqrt{2(\sigma^2 + 2q\mu^2)/\pi} \right|}{\sqrt{2(\sigma^2 + 2q\mu^2)/\pi}} \leq \eta, \tag{14}$$

if $k \geq \left( \frac{4\sigma^3[p+2q(1+\mu^2/\sigma^2)^{3/2}]}{(\sigma^2+2q\mu^2)^{3/2}\sqrt{2\eta}} + \frac{3\sigma^4+2q(6\sigma^2\mu^2+\mu^4)}{(\sigma^2+2q\mu^2)\eta} \right)^2$. As remarked in Theorem 1, the lower bound of $k$ derived in equation 14 for a given small $\eta$ suggests a small matrix sparsity $k$ that performs comparably to larger sparsity values in classification.

The resemblance between the results of Theorems 1 and 2 is not unexpected, as the ternary discrete distribution $x_i \sim \mathcal{T}(\mu, p, q)$ can be regarded as a limiting case of the three-component Gaussian mixture $x_i \sim \mathcal{M}(\mu, \sigma^2, p, q)$, where $\sigma \to 0$. Due to the excellent generalizability of Gaussian mixture models, the two properties discussed above apply to a diverse range of real-world data, as demonstrated in our experiments.

In Appendix C.2, we validate the results of Theorem 2 through numerical computations and statistical simulations. Similarly to the analysis of Theorem 1, we require the matrix size $m \geq \mathcal{O}(\sqrt{n})$, such that the actual distance $\|\boldsymbol{R}^{\top}\boldsymbol{x}\|_1$ for a specific matrix can approximate the expected distance $\mathbb{E}\|\boldsymbol{R}^{\top}\boldsymbol{x}\|_1$ (or equivalently, $m\mathbb{E}|\boldsymbol{r}^{\top}\boldsymbol{x}|$) derived using equation 9. This ensures that the results of Theorem 2 which are based on $\mathbb{E}\|\boldsymbol{R}^{\top}\boldsymbol{x}\|_1$, also apply to $\|\boldsymbol{R}^{\top}\boldsymbol{x}\|_1$. The analysis is similar to that in Property 2, omitted here. By statistical simulations, we also see the similar trends of $\mathrm{MAD}(\boldsymbol{z})$ and $\mathbb{E}|\boldsymbol{r}^{\top}\boldsymbol{x}|$ with respect to variations in $k$, validating the equivalence relation between them as demonstrated in Section 2.3.

## 4 EXPERIMENTS

In this section, we aim to verify that impact of sparse matrices on classification is consistent with our theoretical predictions outlined in Theorems 1 and 2. More precisely, sparse matrices with only one or a few number of nonzero entries per row, can achieve classification performance comparable or even superior to that of denser matrices, particularly under the matrix size $m \geq \mathcal{O}(\sqrt{n})$.

### 4.1 SETTING

For the sake of generality, we evaluate our classification performance across datasets with diverse attributes and scales, including the YaleB image dataset (Georghiades et al., 2001; Lee et al., 2005), the Newsgroups text dataset (Joachims, 1997), the AMLALL gene dataset (Golub et al., 1999), the MNIST binary image dataset (Deng, 2012), the CIFAR100 image dataset (Krizhevsky & Hinton, 2009), as well as the large-scale ImageNet1000 image dataset (Krizhevsky et al., 2012). While most datasets can be approximately modeled using Gaussian mixtures, the MNIST dataset follows two-point distributions. For data details, refer to Appendix D.1.

To clearly reflect the separability of projected data, we adopt the naive $K$-nearest neighbor ($K$NN) classifier (Cover & Hart, 1967), which relies solely on pairwise data similarities without additional discrimination enhancement. For comprehensive validation of our theoretical results, we design two experimental settings: 1) binary classification on classical datasets (Figure 1), to verify basic performance; and 2) multiclass classification on large-scale datasets (Figure 2), specifically CIFAR100 and ImageNet1000, to test scalability under complex scenarios. Similarly to $K$NN, the desired performance trends can also be obtained with other more sophisticated classifiers, like SVMs (Cortes & Vapnik, 1995), as detailed in Appendix D.2. We also include Gaussian random projection as a baseline, given its prevalent use in the random projection research.

### 4.2 RESULTS

As shown in Figures 1 and 2, we evaluate the classification performance of sparse matrices across sparsity levels $k \in [1, 30]$ and projection ratios $m/n \in \{1\%, 10\%, 50\%\}$. The data dimension $n$ is

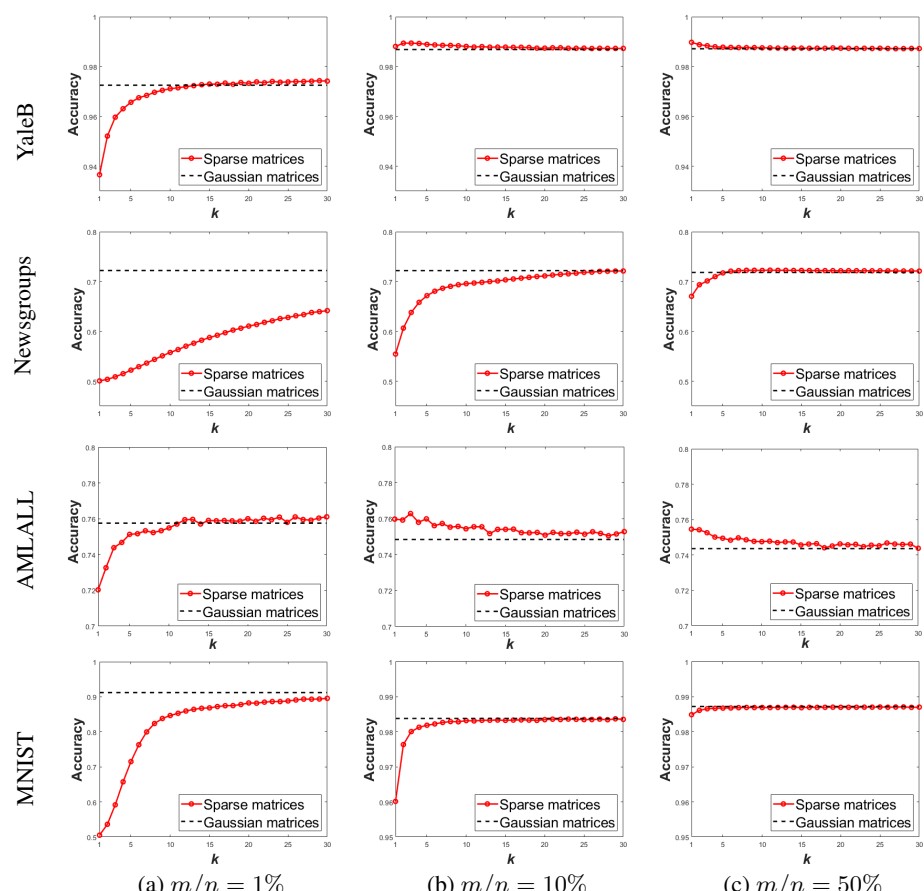

Figure 1: Classification accuracy of sparse matrix-based random projections with varying matrix sparsity $k \in [1, 30]$ and three different projection ratios $m/n = 1\%$, 10% and 50%. The data include the image data (YaleB, DCT features), text data (Newsgroups), gene expression data (AMLALL) and binary data (MNIST, binarized pixels). The performance of Gaussian matrix-based random projections is provided as a baseline.

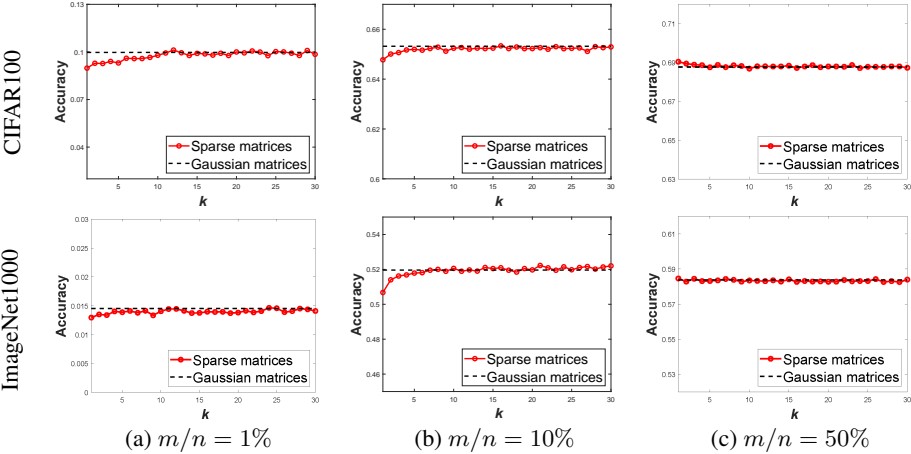

Figure 2: Multiclass classification accuracy of sparse matrix-based random projections with varying matrix sparsity $k \in [1, 30]$ and projection ratio $m/n \in \{1\%, 10\%, 50\%\}$. The datasets include CIFAR100 and ImageNet1000, with features extracted via ViT-B/32 (Dosovitskiy et al., 2020). The performance of Gaussian matrix-based random projections is provided as a baseline.

on the order of thousands. At this scale, the condition $m \geq \mathcal{O}(\sqrt{n})$ is naturally met for $m/n = 10\%$ and $50\%$, but not for $m/n = 1\%$.

These results reveal two main trends. First, classification accuracy stabilizes quickly as the projection ratio $m/n$ increases. Convergence is achieved with relatively small $k$ values ($k < 30$), when $m/n \geq 10\%$. Second, when $m/n \geq 50\%$, the best or near-best classification performance is typically attained at $k = 1$, i.e., when each row contains only one nonzero entry. Overall, the empirical trends in Figures 1 and 2 align closely with the theoretical patterns shown in Figures 3 and 6 (a,b) of Appendix C. These findings strongly support our theoretical conclusion: sparse matrices with only one or very few (i.e. $k < 30$) nonzero entries per row can match or exceed the performance of denser alternatives.

Furthermore, our experiments cover a wide range of datasets, from small ones like YaleB to large-scale ImageNet1000, and include diverse data types such as image features, text features, gene data, and binary-quantized data. This breadth thoroughly validates the universality and robustness of our theoretical insights.

Finally, sparse matrices demonstrate comparable and sometimes superior performance to Gaussian random matrices (the standard baseline), while offering significantly lower computational complexity. This makes them a compelling alternative to Gaussian matrices in real-world applications.

## 5 CONCLUSION

In our analysis of sparse $\{0, \pm 1\}$-matrix-based random projections, we demonstrate that matrices with only one or few (such as less than thirty) nonzero entries per row can achieve comparable or even superior classification performance to denser alternatives. This theoretical result is consistent with classification experiments conducted across various datasets, ranging from small datasets like YaleB to the large dataset ImageNet1000, and encompassing a variety of data types, including image features, text features, gene data, and binary-quantized data. This showcases the broad applicability of our theoretical findings.

Moreover, our experimental results indicate that our extremely sparse matrices perform comparably to, and sometimes even better than, the commonly-used Gaussian matrices. Given the fundamental role and wide applications of random projection in machine learning for dimensionality reduction, the use of our sparse matrices can significantly reduce the computational complexity of related models, such as large-scale retrieval systems Charikar (2002), without compromising accuracy.

Furthermore, our study provides insights into the sparse structures inherent in more advanced models, such as deep networks (Li et al., 2016; Zhu et al., 2017; Wan et al., 2018; Marban et al., 2020; Rokh et al., 2023), where each layer can be modeled using random projections (Giryes et al., 2016).

ACKNOWLEDGMENTS

The language of the paper has been polished by large language models.

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

## A  PROOFS FOR PROPERTY 1 IN SECTION 2

*Proof.* For a $m$-dimensional random vector $\boldsymbol{z}$ with i.i.d. entries $z_i \sim \mathcal{N}(\mu, \sigma^2)$, we have

$$\mathbb{E}\|\boldsymbol{z} - \mathbb{E}\boldsymbol{z}\|_1 = \mathbb{E}\left(\sum_{i=1}^{m} |z_i - \mathbb{E}z_i|\right) = \sum_{i=1}^{m} \mathbb{E}|z_i - \mathbb{E}z_i| = m\sqrt{\frac{2}{\pi}}\sigma \tag{15}$$

and

$$\mathbb{E}\|\boldsymbol{z} - \boldsymbol{z}'\|_1 = \mathbb{E}\left(\sum_{i=1}^{m} |z_i - z'_i|\right) = \sum_{i=1}^{m} \mathbb{E}|z_i - z'_i| = m\frac{2}{\sqrt{\pi}}\sigma \tag{16}$$

where $\boldsymbol{z}'$ is an independent copy of $z$. Combining the above two results leads to

$$\mathbb{E}\|\boldsymbol{z} - \mathbb{E}\boldsymbol{z}\|_1 = \frac{1}{\sqrt{2}}\mathbb{E}\|\boldsymbol{z} - \boldsymbol{z}'\|_1 \tag{17}$$

$\square$

## B  PROOFS FOR THEOREMS 1-2 AND PROPERTY 2 IN SECTION 3

### B.1  PROOF OF THEOREM 1

*Proof.* In the following, we sequentially prove equation 4, equation 5, P1 and P2.

**Proofs of equation 4 and equation 5:** With the distributions of $\boldsymbol{r}$ and $\boldsymbol{x}$, we can write $\|\boldsymbol{r}^\top \boldsymbol{x}\|_1 = \sqrt{\frac{n}{mk}}\mu \left|\sum_{i=1}^{k} z_i\right|$, where $z_i \in \{-1, 0, 1\}$ with probabilities $\{q, p, q\}$. Then, it can be derived that

$$\mathbb{E}|\boldsymbol{r}^\top \boldsymbol{x}| = \mu\sqrt{\frac{n}{mk}} \sum_{i=0}^{k} C_k^i p^i q^{k-i} \sum_{j=0}^{k-i} C_{k-i}^j |k - i - 2j|, \tag{18}$$

among which $\sum_{j=0}^{k-i} C_{k-i}^j |k - i - 2j|$ can be expressed as

$$\sum_{j=0}^{k-i} (C_{k-i}^j |k - i - 2j|) = 2\left\lceil \frac{k-i}{2} \right\rceil C_{k-i}^{\lceil \frac{k-i}{2} \rceil}, \tag{19}$$

where $\lceil \alpha \rceil = \min\{\beta : \beta \geq \alpha, \beta \in \mathbb{Z}\}$. Combine (18) and (19), we can obtain equation 4.

Next, we can derive the variance of $|\boldsymbol{r}^\top \boldsymbol{x}|$

$$\begin{aligned} \mathrm{Var}(|\boldsymbol{r}^\top \boldsymbol{x}|) &= \mathrm{Var}(\boldsymbol{r}^\top \boldsymbol{x}) - \left(\mathbb{E}|\boldsymbol{r}^\top \boldsymbol{x}|\right)^2 \\ &= \frac{2q\mu^2 n}{m} - \frac{4\mu^2 n}{mk}\left(\sum_{i=0}^{k} C_k^i p^i q^{k-i} \left\lceil \frac{k-i}{2} \right\rceil C_{k-i}^{\lceil \frac{k-i}{2} \rceil}\right)^2. \end{aligned} \tag{20}$$

**Proof of P1:** This part aims to prove

$$\mathbb{E}|\boldsymbol{r}^\top \boldsymbol{x}|_{k=1} > \mathbb{E}|\boldsymbol{r}^\top \boldsymbol{x}|_{k>1},$$

where the subscript $k = 1$ denotes the case of $\mathbb{E}|\boldsymbol{r}^\top \boldsymbol{x}|$ with $k = 1$, and the subscript $k > 1$ means the case of $k$ taking any integer value greater than 1. In the following, we will calculate and compare $\mathbb{E}|\boldsymbol{r}^\top \boldsymbol{x}|$ in terms of the two cases. For the case of $k = 1$, by equation 4, it is easy to derive that

$$\mathbb{E}|\boldsymbol{r}^\top \boldsymbol{x}|_{k=1} = 2q\mu\sqrt{\frac{n}{m}}. \tag{21}$$

Then, let us see the case of computing $\mathbb{E}|\boldsymbol{r}^\top\boldsymbol{x}|_{k>1}$. By equation 4, $\mathbb{E}|\boldsymbol{r}^\top\boldsymbol{x}|_{k>1}$ is the sum of $\frac{2}{\sqrt{k}}C_k^i p^i q^{k-i}\left\lceil\frac{k-i}{2}\right\rceil C_{k-i}^{\lceil\frac{k-i}{2}\rceil}$ multiplied by $\mu\sqrt{\frac{n}{m}}$. To compute $\frac{2}{\sqrt{k}}C_k^i p^i q^{k-i}\left\lceil\frac{k-i}{2}\right\rceil C_{k-i}^{\lceil\frac{k-i}{2}\rceil}$, we consider separately two cases: $k-i$ is even or odd, as detailed below.

**Case 1:** Suppose $k-i$ is even. We have

$$
\frac{2}{\sqrt{k}}C_k^i p^i q^{k-i}\left\lceil\frac{k-i}{2}\right\rceil C_{k-i}^{\lceil\frac{k-i}{2}\rceil}
$$
$$
\leq \frac{1}{\sqrt{k}}C_k^i p^i q^{k-i}(k-i)2^{k-i}\sqrt{\frac{2}{(k-i)\pi}}
$$
$$
\leq \sqrt{\frac{2}{\pi}}C_k^i p^i (2q)^{k-i}, \tag{22}
$$

since $C_{2\gamma}^\gamma \leq \frac{2^{2\gamma}}{\sqrt{\gamma\pi}}$, where $\gamma$ is a positive integer (Stănică, 2001).

**Case 2:** Suppose $k-i$ is odd. We have

$$
\frac{2}{\sqrt{k}}C_k^i p^i q^{k-i}\left\lceil\frac{k-i}{2}\right\rceil C_{k-i}^{\lceil\frac{k-i}{2}\rceil}
$$
$$
\leq \frac{1}{\sqrt{k}}C_k^i p^i q^{k-i}(k-i)2^{k-i}\sqrt{\frac{2}{(k-i-1)\pi}}
$$
$$
= \sqrt{\frac{2}{\pi}}C_k^i p^i (2q)^{k-i}\frac{k-i}{\sqrt{k(k-i-1)}} \tag{23}
$$

Given $k \geq 5$, we further have

$$
\frac{k-i}{\sqrt{k(k-i-1)}} < 1 \ \text{ for } \ 2 \leq i \leq k-2,
$$

and for $i = k-1$ or $k$,

$$
\frac{2}{\sqrt{k}}C_k^i p^i q^{k-i}\left\lceil\frac{k-i}{2}\right\rceil C_{k-i}^{\lceil\frac{k-i}{2}\rceil} < \sqrt{\frac{2}{\pi}}C_k^i p^i (2q)^{k-i}.
$$

To sum up, when $k-i$ is odd,

$$
\frac{2}{\sqrt{k}}C_k^i p^i q^{k-i}\left\lceil\frac{k-i}{2}\right\rceil C_{k-i}^{\lceil\frac{k-i}{2}\rceil}
$$
$$
\leq \begin{cases} \sqrt{\dfrac{2}{\pi}}C_k^i p^i (2q)^{k-i}, & k \geq 5, i \geq 2, \\ \dfrac{2}{\sqrt{k}}C_k^i p^i q^{k-i}(k-i)C_{k-i-1}^{\frac{k-i-1}{2}}, & \text{otherwise.} \end{cases} \tag{24}
$$

According to the results equation 22 and equation 24 derived in the above two cases, we know that $\mathbb{E}|\boldsymbol{r}^\top\boldsymbol{x}|_{k>1}$ can be computed in terms of two cases, $2 \leq k \leq 4$ and $k \geq 5$. For the case of $2 \leq k \leq 4$, by equation 4, we have

$$
\mathbb{E}|\boldsymbol{r}^\top\boldsymbol{x}| = \begin{cases} \dfrac{\mu\sqrt{n}}{\sqrt{2m}}(4q^2 + 4pq), & k = 2, \\[2mm] \dfrac{\mu\sqrt{n}}{\sqrt{3m}}(12q^3 + 12pq^2 + 6p^2 q), & k = 3, \\[2mm] \dfrac{\mu\sqrt{n}}{\sqrt{m}}(12q^4 + 24pq^3 + 12p^2 q^2 + 4p^3 q), & k = 4, \end{cases} \tag{25}
$$

and for the case of $k \geq 5$, with equation 22 and equation 24, we have

$$\mathbb{E}|\boldsymbol{r}^\top \boldsymbol{x}| \leq \mu\sqrt{\frac{2n}{\pi m}} + \mu\sqrt{\frac{n}{m}}(2q)^5\left(\frac{3\sqrt{5}}{8} - \sqrt{\frac{2}{\pi}}\right). \tag{26}$$

By equation 21, equation 25 and equation 26, we can derive that

$$\mathbb{E}|\boldsymbol{r}^\top \boldsymbol{x}|_{k=1} > \mathbb{E}|\boldsymbol{r}^\top \boldsymbol{x}|_{k>1}$$

holds under the condition of $p \leq 0.188$. Then P1 is proved.

In what follows, we elaborate the proof of equation 26 by considering two cases of $k$, being even or odd.

**Case 1:** Suppose $k \geq 5$ and $k$ is even. Combining equation 22 and equation 24, we have

$$\mathbb{E}|\boldsymbol{r}^\top \boldsymbol{x}| \leq \mu\sqrt{\frac{n}{m}}C_k^1 p(2q)^{k-1}\left(\frac{\sqrt{k}}{2^{k-1}}C_{k-1}^{\frac{k}{2}-1} - \sqrt{\frac{2}{\pi}}\right)$$
$$+ \mu\sqrt{\frac{2n}{\pi m}}\sum_{i=0}^{k}C_k^i p^i(2q)^{k-i}. \tag{27}$$

Denote $h_1(k) = \frac{\sqrt{k}}{2^{k-1}}C_{k-1}^{\frac{k}{2}-1}$. For

$$\frac{h_1(k+2)}{h_1(k)} = \frac{k+1}{\sqrt{k(k+2)}} > 1$$

we have

$$h_1(k) = \frac{\sqrt{k}}{2^{k-1}}C_{k-1}^{\frac{k}{2}-1} \leq \lim_{k\to\infty} h_1(k) = \sqrt{\frac{2}{\pi}}. \tag{28}$$

Then, it follows from (27) and (28) that

$$\mathbb{E}|\boldsymbol{r}^\top \boldsymbol{x}| \leq \mu\sqrt{\frac{2n}{\pi m}}. \tag{29}$$

**Case 2:** Suppose $k \geq 5$ and $k$ is odd. Combining (22) and (24), we have

$$\mathbb{E}|\boldsymbol{r}^\top \boldsymbol{x}| \leq \mu\sqrt{\frac{n}{m}}C_k^0(2q)^k\left(\frac{\sqrt{k}}{2^{k-1}}C_{k-1}^{\frac{k-1}{2}} - \sqrt{\frac{2}{\pi}}\right)$$
$$+ \mu\sqrt{\frac{2n}{\pi m}}\sum_{i=0}^{k}C_k^i p^i(2q)^{k-i}. \tag{30}$$

Denote $h_2(k) = \frac{\sqrt{k}}{2^{k-1}}C_{k-1}^{\frac{k-1}{2}}$. For

$$\frac{h_2(k+2)}{h_2(k)} = \frac{\sqrt{k(k+2)}}{k+1} < 1$$

we have

$$h_2(k) = \frac{\sqrt{k}}{2^{k-1}}C_{k-1}^{\frac{k-1}{2}} \leq h_2(5) = \frac{\sqrt{5}}{2^4}C_4^2. \tag{31}$$

Then, it follows from (30) and (31) that

$$\mathbb{E}|\boldsymbol{r}^\top \boldsymbol{x}| \leq \mu\sqrt{\frac{2n}{\pi m}} + \mu\sqrt{\frac{n}{m}}(2q)^5\left(\frac{3\sqrt{5}}{8} - \sqrt{\frac{2}{\pi}}\right).$$

**Proof of P2:** For ease of analysis, we first define the function

$$g(\boldsymbol{r}^\top \boldsymbol{x}; k, p) = \frac{\mathbb{E}|\boldsymbol{r}^\top \boldsymbol{x}|_k}{\mu\sqrt{n/m}} = \mathbb{E}\left|\frac{1}{\sqrt{k}}\sum_{i=1}^{k} z_i\right|, \tag{32}$$

where $\{z_i\}$ is independently and identically distributed and $z_i \in \{-1, 0, 1\}$ with probabilities $\{q, p, q\}$. By the Lindeberg-Lévy Central Limit Theorem, we have

$$\frac{1}{\sqrt{k}}\sum_{i=1}^{k} z_i \rightsquigarrow Z, \tag{33}$$

where $Z \sim N(0, 2q)$.

Then based on equation 26, we have for $k \geq 5$,

$$\mathbb{E}\left|\frac{1}{\sqrt{k}}\sum_{i=1}^{k} z_i\right| \leq \sqrt{\frac{2}{\pi}} + (2q)^5\left(\frac{3\sqrt{5}}{8} - \sqrt{\frac{2}{\pi}}\right).$$

It means that

$$\lim_{M\to+\infty}\limsup_{k\to+\infty}\mathbb{E}\left[\left|\frac{1}{\sqrt{k}}\sum_{i=1}^{k} z_i\right|\mathbb{1}\left\{\left|\frac{1}{\sqrt{k}}\sum_{i=1}^{k} z_i\right| > M\right\}\right] = 0.$$

Hence, $\left|\frac{1}{\sqrt{k}}\sum_{i=1}^{k} z_i\right|$ is an asymptotically uniformly integrable sequence.

According to Theorem 2.20 in (Van der Vaart, 2000), we obtain

$$\lim_{k\to+\infty}\frac{\sqrt{m}}{\mu\sqrt{n}}\mathbb{E}|\boldsymbol{r}^\top \boldsymbol{x}| = \lim_{k\to+\infty}\mathbb{E}\left|\frac{1}{\sqrt{k}}\sum_{i=1}^{k} z_i\right|$$

$$= \mathbb{E}|Z|$$

$$= 2\sqrt{\frac{q}{\pi}}.$$

Next, let us investigate the error of the above convergence with respect to $k$. Following the definitions and properties described in equation 32 and equation 33, we further suppose $t_i = \frac{1}{\sqrt{2q}}z_i$ and $Q \sim N(0, 1)$, and get

$$\left|\frac{\sqrt{m}}{\mu\sqrt{n}}\mathbb{E}|\boldsymbol{r}^\top \boldsymbol{x}| - 2\sqrt{q/\pi}\right|$$

$$= \left|\mathbb{E}\left|\frac{1}{k}\sum_{i=1}^{k} z_i\right| - \mathbb{E}|Z|\right|$$

$$= \sqrt{2q}\left|\mathbb{E}\left|\frac{1}{k}\sum_{i=1}^{k} t_i\right| - \mathbb{E}|Q|\right|$$

$$\leq \sqrt{2q}d_w\left(\mathbb{E}\left|\frac{1}{k}\sum_{i=1}^{k} t_i\right|, \mathbb{E}|Q|\right)$$

where $d_w(\nu, \upsilon)$ denotes the Kolmogorov metric:

$$d_w(\nu, \upsilon) = \sup_{h\in\mathcal{H}}\left|\int h(x)d\nu(x) - \int h(x)d\upsilon(x)\right|,$$

$$\mathcal{H} = \{h : \mathbb{R} \to \mathbb{R} : |h(x) - h(y)| \leq |x - y|\}.$$

By the Theorem 3.2 in (Ross, 2011), since $\{t_i\}$ are i.i.d and $\mathbb{E}t_i = 0$, $\mathbb{E}t_i^2 = 1$, $\mathbb{E}|t_i|^4 < \infty$, we have

$$d_w\left(\mathbb{E}\left|\frac{1}{k}\sum_{i=1}^{k} t_i\right|, \mathbb{E}|Q|\right) \leq \frac{1}{k^{3/2}}\sum_{i=1}^{k}\mathbb{E}|t_i|^3 + \frac{\sqrt{2}}{\sqrt{\pi}k}\sqrt{\sum_{i=1}^{k}\mathbb{E}t_i^4}$$

$$= \frac{1}{\sqrt{2qk}} + \frac{\sqrt{2}}{\sqrt{2q\pi k}},$$

and then

$$\left| \frac{\sqrt{m}}{\mu\sqrt{n}} \mathbb{E}|\boldsymbol{r}^\top \boldsymbol{x}| - 2\sqrt{q/\pi} \right| \le \frac{\sqrt{\pi} + \sqrt{2}}{\sqrt{\pi k}}.$$

$\square$

## B.2 PROOF OF PROPERTY 2

*Proof.* This problem can be addressed using the Chebyshev's Inequality, which requires us to first derive $\mathbb{E}z$ and $\mathrm{Var}(z)$. Note that $\mathbb{E}z = \mathbb{E}(\frac{1}{m}\sum_{i=1}^m |\boldsymbol{r}_i^\top \boldsymbol{x}|) = \mathbb{E}(|\boldsymbol{r}_i^\top \boldsymbol{x}|)$ has been derived in equation 4. In the sequel, we need to first solve $\mathrm{Var}(z) = \mathbb{E}z^2 - (\mathbb{E}z)^2$, which has

$$\begin{aligned}
\mathbb{E}z^2 &= \mathbb{E}(\frac{1}{m}\sum_{i=1}^m |\boldsymbol{r}_i^\top \boldsymbol{x}|)^2 \\
&= \frac{1}{m^2}\mathbb{E}(\sum_{i=1}^m |\boldsymbol{r}_i^\top \boldsymbol{x}|^2) + \frac{1}{m^2}\mathbb{E}(\sum_{i \ne j} |\boldsymbol{r}_i^\top \boldsymbol{x}| \cdot |\boldsymbol{r}_j^\top \boldsymbol{x}|) \\
&= \frac{2q\mu^2 n}{m^2} + \frac{m-1}{2m}\mathbb{E}(|\boldsymbol{r}_i^\top \boldsymbol{x}| \cdot |\boldsymbol{r}_j^\top \boldsymbol{x}|).
\end{aligned} \tag{34}$$

For the second term in the above result, it holds

$$\mathbb{E}(|\boldsymbol{r}_i^\top \boldsymbol{x}| \cdot |\boldsymbol{r}_j^\top \boldsymbol{x}|) \le \mathrm{Var}(|\boldsymbol{r}_i^\top \boldsymbol{x}|) + (\mathbb{E}|\boldsymbol{r}_i^\top \boldsymbol{x}|)^2 = \mathrm{Var}(|\boldsymbol{r}_i^\top \boldsymbol{x}|) + (\mathbb{E}z)^2, \tag{35}$$

by the covariance property

$$\begin{aligned}
\mathrm{Cov}(|\boldsymbol{r}_i^\top \boldsymbol{x}|, |\boldsymbol{r}_j^\top \boldsymbol{x}|) &= \mathbb{E}(|\boldsymbol{r}_i^\top \boldsymbol{x}| \cdot |\boldsymbol{r}_j^\top \boldsymbol{x}|) - \mathbb{E}|\boldsymbol{r}_i^\top \boldsymbol{x}| \cdot \mathbb{E}|\boldsymbol{r}_j^\top \boldsymbol{x}| \\
&= \rho\sqrt{\mathrm{Var}(|\boldsymbol{r}_i^\top \boldsymbol{x}|)} \cdot \sqrt{\mathrm{Var}(|\boldsymbol{r}_j^\top \boldsymbol{x}|)} \\
&= \rho\mathrm{Var}(|\boldsymbol{r}_i^\top \boldsymbol{x}|),
\end{aligned} \tag{36}$$

where $\rho \in (-1, 1)$ is the correlation coefficient.

Substituting equation 34 into $\mathrm{Var}(z) = \mathbb{E}z^2 - (\mathbb{E}z)^2$, by the inequality equation 35 and equation 20, we can derive

$$\begin{aligned}
\mathrm{Var}(z) &\le \frac{2q\mu^2 n}{m^2} + \frac{m-1}{2m}[\mathrm{Var}(|\boldsymbol{r}_i^\top \boldsymbol{x}|) + (\mathbb{E}z)^2] - (\mathbb{E}z)^2 \\
&= \frac{2q\mu^2 n}{m^2} + \frac{m-1}{2m} \cdot \frac{2q\mu^2 n}{m^2} - (\mathbb{E}z)^2 \\
&= \frac{(m+1)q\mu^2 n}{m^2} - (\mathbb{E}z)^2.
\end{aligned} \tag{37}$$

With the above inequality about $\mathrm{Var}(z)$, we can further explore the condition that holds the desired probability

$$\Pr\{|z - \mathbb{E}z| \le \varepsilon\} \ge 1 - \delta. \tag{38}$$

By the Chebyshev's Inequality, equation 38 will be achieved, if $\mathrm{Var}(z)/\varepsilon^2 \le \delta$; and according to equation 37, this condition can be satisfied when $\frac{m^2}{m+1} \ge \frac{q\mu^2 n}{\varepsilon^2 \delta}$.

In the above analysis, we consider a random $\boldsymbol{x}$. For a given $\boldsymbol{x}$, the condition of holding equation 38 can be further relaxed to $m^2 \ge \frac{2q\mu^2 n}{\varepsilon^2 \delta}$, since in this case $|\boldsymbol{r}_i^\top \boldsymbol{x}|$ is independent between different $i \in [m]$, such that $\mathrm{Var}(z)$ changes to be equation 20 divided by $m$. $\square$

## B.3 PROOF OF THEOREM 2

*Proof.* First, we derive the absolute moment of $z \sim \mathcal{N}(\mu, \sigma^2)$ as

$$\mathbb{E}|z| = \sqrt{\frac{2}{\pi}}\sigma e^{-\frac{\mu^2}{2\sigma^2}} + \mu\left(1 - 2\Phi\left(-\frac{\mu}{\sigma}\right)\right) \tag{39}$$

which will be used in the sequel. With the distributions of $r$ and $x$, we have $|r^\top x| = \sqrt{\frac{n}{mk}}\left|\sum_{i=1}^{k} x_i\right|$. For easier expression, assume $y = \sum_{i=1}^{k} x_i$, then the distribution of $y$ can be expressed as

$$f(y) = \sum_{i=0}^{k}\sum_{j=0}^{k-i} C_k^i C_{k-i}^j p^i q^{k-i} \frac{1}{\sqrt{2\pi k}\sigma} e^{-\frac{(y-(2j+i-s)\mu)^2}{2k\sigma^2}}.$$

Then, by equation 39 we can derive that

$$
\begin{aligned}
\mathbb{E}|r^\top x| &= \sqrt{\frac{n}{mk}} \sum_{i=0}^{k}\sum_{j=0}^{k-i} \Bigg[ C_k^i C_{k-i}^j p^i q^{k-i} \\
&\quad \times \int_{-\infty}^{+\infty} \frac{|y|}{\sqrt{2\pi k}\sigma} e^{-\frac{(y-(2j+i-s)\mu)^2}{2k\sigma^2}} dy \Bigg] \\
&= 2\mu\sqrt{\frac{n}{mk}} \sum_{i=0}^{k} C_k^i p^i q^{k-i} \left\lceil \frac{k-i}{2} \right\rceil C_{k-i}^{\left\lceil \frac{k-i}{2} \right\rceil} \\
&\quad - 2\mu\sqrt{\frac{n}{mk}} \sum_{i=0}^{k} C_k^i p^i q^{k-i} \sum_{j=0}^{k-i} C_{k-i}^j \Phi\left( -\frac{|k-i-2j|\mu}{\sqrt{k}\sigma} \right) \\
&\quad + \sigma\sqrt{\frac{2n}{\pi m}} \sum_{i=0}^{k} C_k^i p^i q^{k-i} \sum_{j=0}^{k-i} C_{k-i}^j e^{-\frac{(k-i-2j)^2\mu^2}{2k\sigma^2}}
\end{aligned}
$$

where $\Phi(\cdot)$ is the distribution function of $\mathcal{N}(0,1)$.

The above equation and equation 21, equation 25, equation 26 together lead to

$$\mathbb{E}|r^\top x| \leq \mu\sqrt{\frac{n}{m}} + \sigma\sqrt{\frac{2n}{\pi m}}.$$

Next, we can derive the variance of $|r^\top x|$ as

$$
\begin{aligned}
\mathrm{Var}(|r^\top x|) &= Var(r^\top x) - \left( \mathbb{E}|r^\top x| \right)^2 \\
&= \frac{n}{m}(\sigma^2 + 2q\mu^2) - \left( \mathbb{E}\|r^\top x\|_1 \right)^2.
\end{aligned}
$$

Finally, the convergence of $\frac{\sqrt{m}}{\mu\sqrt{n}}\mathbb{E}|r^\top x|$ shown in equation 12 and equation 13 can be derived in a similar way to the proof of P2 in Theorem 1. □

## C    Numerical validations of Theorems 1 and 2

### C.1    Numerical validation of Theorem 1

#### C.1.1    Validating P1 and P2 by directly computing $\mathbb{E}|\mathbf{r}^{\top}\mathbf{x}|/(\mu\sqrt{n/m})$ with equation 4

To more accurately examine the changing trend of $\mathbb{E}|\boldsymbol{r}^{\top}\boldsymbol{x}|/(\mu\sqrt{n/m})$ against varying matrix sparsity $k$ (derived in P1 and P2), we directly compute the value of $\mathbb{E}|\boldsymbol{r}^{\top}\boldsymbol{x}|/(\mu\sqrt{n/m})$ by equation 4. Note that besides the parameter $k$, $\mathbb{E}|\boldsymbol{r}^{\top}\boldsymbol{x}|/(\mu\sqrt{n/m})$ also involves the parameter $p$, the probability of $x_i = 0$ as specified in equation 1. So we investigate $\mathbb{E}|\boldsymbol{r}^{\top}\boldsymbol{x}|/(\mu\sqrt{n/m})$ over $k \in [1, 500]$ for different $p \in (0, 1)$. For brevity, we here only provide the results of $p = 1/3$ and $2/3$ in Figures 3 (a) and (b). The results exhibit two properties similar to those predicted by P1 and P2:

(P3) When $p \leq 1/3$, such as the case of $p = 1/3$ shown in Figure 3(a), $\mathbb{E}|\boldsymbol{r}^{\top}\boldsymbol{x}|/(\mu\sqrt{n/m})$ tends to achieve its maximum value at $k = 1$, but at other larger $k$ when $p > 1/3$, such as the case of $p = 2/3$ illustrated in Figure 3(b). The results indicate that to maximize $\mathbb{E}|\boldsymbol{r}^{\top}\boldsymbol{x}|/(\mu\sqrt{n/m})$ at $k = 1$, the condition $p \in [0, 1/3)$ is sufficient, which is broader than the theoretical requirement $p \in [0, 0.188)$ derived from P1. Recall that a wider range of $p$ allows for a larger space of data as modeled in (3). This suggests that the desired property of maximizing $\mathbb{E}|\boldsymbol{r}^{\top}\boldsymbol{x}|/(\mu\sqrt{n/m})$ at $k = 1$ can be achieved over a wider range of $p$ values than what was theoretically predicted. To achieve a small $p$ within the range of $p \in [0, 1/3)$, as pointed out in Section 2.2.2, the original data points $\boldsymbol{h}$ and $\boldsymbol{h}'$ need to exhibit sufficiently high discrimination between them.

(P4) With the increasing of $k$, as the two cases of $p = 1/3$ and $2/3$ shown in Figures 3(a) and (b), $\mathbb{E}|\boldsymbol{r}^{\top}\boldsymbol{x}|/(\mu\sqrt{n/m})$ tends to converge to the limit value $2\sqrt{q/\pi}$ derived in equation 6, where $q = (1 - p)/2$. Furthermore, it can be seen that small convergence errors will be achieved, when $k$ is very small, typically in the range of a few tens. For instance, in Figure 4(a) we derive the convergence error ratios as defined in equation 8, which give the values close to zero when $k \geq 20$ and $p$ is relatively small. Recall that the small $p$ value implies that the original data have high discrimination between each other. With the decreasing of data discrimination, we should need larger $k$ to achieve small convergence errors.

In the analysis of the expected distance $\mathbb{E}|\boldsymbol{r}^{\top}\boldsymbol{x}|$, the influence of the variance $\mathrm{Var}(|\boldsymbol{r}^{\top}\boldsymbol{x}|)$ in equation 5 should be considered. Statistically, a lower variance $\mathrm{Var}(|\boldsymbol{r}^{\top}\boldsymbol{x}|)$ indicates a higher probability that the actual distance $|\boldsymbol{r}^{\top}\boldsymbol{x}|$ of a single matrix closely approximates its expected value $\mathbb{E}|\boldsymbol{r}^{\top}\boldsymbol{x}|$. Also, this implies a higher consistence between theoretical and practical results. By computing equation 5, we observe a trend similar to $\mathbb{E}|\boldsymbol{r}^{\top}\boldsymbol{x}|$: as $k$ increases, $\mathrm{Var}(|\boldsymbol{r}^{\top}\boldsymbol{x}|)$ tends to quickly converge to a constant value. This suggests that $\mathrm{Var}(|\boldsymbol{r}^{\top}\boldsymbol{x}|)$ varies minimally across different $k$ values. Therefore, the probability of $|\boldsymbol{r}^{\top}\boldsymbol{x}|$ approximating $\mathbb{E}|\boldsymbol{r}^{\top}\boldsymbol{x}|$ remains consistent for various $k$, enabling us to use $\mathbb{E}|\boldsymbol{r}^{\top}\boldsymbol{x}|$ to reasonably estimate and compare the distances $|\boldsymbol{r}^{\top}\boldsymbol{x}|$ of actual matrices across different $k$.

#### C.1.2    Validating P1 and P2 by statistically estimating $\mathbb{E}|\mathbf{r}^{\top}\mathbf{x}|/(\mu\sqrt{n/m})$ with synthetic data

To verify the correctness of Theorem 1, including the expression equation 4 of $\mathbb{E}|\boldsymbol{r}^{\top}\boldsymbol{x}|$ and its two properties P1 and P2, we here estimate the expectation value $\mathbb{E}|\boldsymbol{r}^{\top}\boldsymbol{x}|/(\mu\sqrt{n/m})$ (against varying $k$) by averaging over the statistically generated samples of $\boldsymbol{r}$ and $\boldsymbol{x}$. If the theorem results are correct, the statistical simulation results should be consistent with the numerical computation results P3 and P4 (derived by Theorem 1). The simulation is introduced as follows. First, we randomly generate $10^6$ pairs of $\boldsymbol{r}$ and $\boldsymbol{x}$ from their respective distributions, i.e. $\boldsymbol{r} \in \{0, \pm\sqrt{\frac{n}{mk}}\}^n$ with $k$ nonzero entries randomly distributed, and $\boldsymbol{x}$ with i.i.d. $x_i \sim \mathcal{T}(\mu, p, q)$. Then, the average value of $|\boldsymbol{r}^{\top}\boldsymbol{x}|/(\mu\sqrt{n/m})$ is derived as the final estimate of $\mathbb{E}|\boldsymbol{r}^{\top}\boldsymbol{x}|/(\mu\sqrt{n/m})$. The parameters for the distributions of $\boldsymbol{r}$ and $\boldsymbol{x}$ are set as follows: $m = 1$, $n = 10^4$, $\mu = 1$, and $p = 1/3$ or $2/3$. The data dimension $n = 10^4$ allows us to increase $k$ from 1 to $10^4$. The average value of $|\boldsymbol{r}^{\top}\boldsymbol{x}|/(\mu\sqrt{n/m})$ at each $k$ is provided in Figures 3(c) and (d), respectively for the cases of $p = 1/3$ and $2/3$. Note

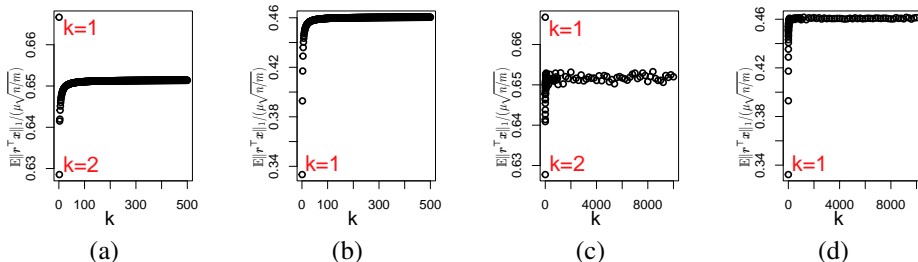

Figure 3: The value of $\mathbb{E}|\boldsymbol{r}^\top \boldsymbol{x}|/(\mu\sqrt{n/m})$ calculated by equation 4 with $p = 1/3$ (a) and $p = 2/3$ (b), and estimated by statistical simulation with $p = 1/3$ (c) and $p = 2/3$ (d), provided $x_i \sim \mathcal{T}(\mu, p, q), \mu = 1$.

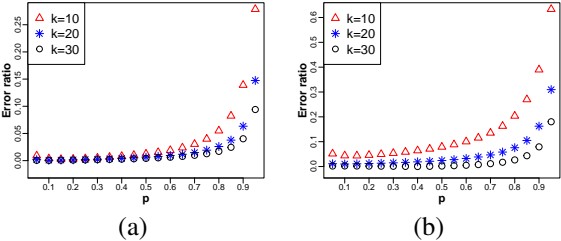

Figure 4: The convergence error ratios of three different $k \in \{10, 20, 30\}$ over varying $p$ are derived for two-point distributed data (a) and Gaussian mixture data (b), by computing the left side of the inequality of $\eta$ shown respectively in equation 8 and equation 14.

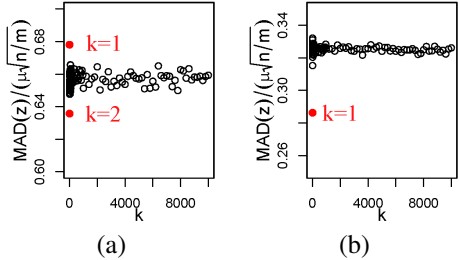

Figure 5: The value of $\mathrm{MAD}(\boldsymbol{z})$ against varying $k$ is estimated in (a) and (b), respectively, with the synthetic data generated using the parameters $p = 1/3$ and $p = 2/3$ as defined in Figures 3 (c) and (d) for $x_i \sim \mathcal{T}(\mu, p, q), \mu = 1$.

that the choices of $m$, $n$ and $\mu$ will not affect the changing trend of $\mathbb{E}|\boldsymbol{r}^\top \boldsymbol{x}|/(\mu\sqrt{n/m})$ against $k$. Comparing the numerical computation results with the simulation results presented in Figure 3, specifically contrasting (a) vs. (c) and (b) vs. (d), it is seen that both sets of results exhibit similar trends in the variation of $\mathbb{E}|\boldsymbol{r}^\top \boldsymbol{x}|/(\mu\sqrt{n/m})$. The similarity between them validates Theorem 1, as well as the numerical computation results P3 and P4.

### C.1.3 VALIDATING EQUIVALENCE RELATIONS BETWEEN $\mathrm{MAD}(z)$ AND $\mathbb{E}|r^\top x|$

In Section 2.3, our statistical analysis demonstrates that the trend of $\mathrm{MAD}(\boldsymbol{z})$ against varying matrix sparsity $k$ can be estimated with $\mathbb{E}|\boldsymbol{r}^\top \boldsymbol{x}|$. By comparing the simulation values of $\mathrm{MAD}(\boldsymbol{z})$ (Figures 5 (a) and (b)) with those of $\mathbb{E}|\boldsymbol{r}^\top \boldsymbol{x}|$ (Figures 3 (c) and (d)), it is seen that they indeed exhibit similar trends. This validates the equivalence relation between them.

## C.2 NUMERICAL VALIDATION OF THEOREM 2

### C.2.1 VALIDATING EQUATION 12 BY DIRECTLY COMPUTING $\mathbb{E}|\mathbf{r}^\top\mathbf{x}|/(\mu\sqrt{n/m})$ WITH EQUATION 9

In this part, we directly compute the value of $\mathbb{E}|\mathbf{r}^\top\mathbf{x}|/(\mu\sqrt{n/m})$ by equation 9. Note that $\mathbb{E}|\mathbf{r}^\top\mathbf{x}|/(\mu\sqrt{n/m})$ involves four parameters: $k$, $p$, $\mu$, and $\sigma$. In computing equation 9, we fix $\mu = 1$ and vary other parameters in the ranges of $\sigma/\mu \in (0, 1/3)$, $p \in (0, 1)$ and $k \in [1, 500]$. Here, we upper bound the value range of $\sigma/\mu$ by $1/3$ for easy simulation. Empirically, the changing trend of $\mathbb{E}|\mathbf{r}^\top\mathbf{x}|/(\mu\sqrt{n/m})$ is not sensitive to the varying of $\sigma/\mu$, but sensitive to $p$, i.e. the probability of each entry $x_i$ of the data difference $\mathbf{x}$ taking zero value, as specified in equation 2. In Figures 6(a) and (b), we provide two typical results of $p = 1/2$ and $2/3$, and observe two properties similar to the previous P3 and P4:

(P5) When $p \leq 1/2$, such as the case of $p = 1/2$ and $\sigma/\mu = 1/3$ shown in Figure 6(a), $\mathbb{E}|\mathbf{r}^\top\mathbf{x}|/\mu\sqrt{n/m}$ tends to obtain its maximum at $k = 1$, but at other larger $k$ when $p > 1/2$, such as the case of $p = 2/3$ and $\sigma/\mu = 1/3$ shown in Figure 6(b). It can be seen that the upper bound of $p$ obtained here for Gaussian mixture data is relaxed from 2/3 to 1/2 compared to the bound derived in P3 for two-point distributed data. This implies a wider range of data distributions that enable obtaining the maximum $\mathbb{E}|\mathbf{r}^\top\mathbf{x}|/\mu\sqrt{n/m}$ at $k = 1$.

(P6) With the increasing of $k$, as the two results shown in Figure 6(a) and (b), $\mathbb{E}|\mathbf{r}^\top\mathbf{x}|/(\mu\sqrt{n/m})$ converges to the limit value derived in equation 12. Similarly to the convergence discussed in P4 for two-point distributed data, the convergence error ratio defined in equation 14 can approach zero with small $k$, such as $k = 20$ shown in Figure 4(b), especially when $p$ is relatively small, namely the original data having relatively high discrimination.

For P5 and P6, their similarity to P3 and P4 is not surprising, since the ternary discrete distribution $x_i \sim \mathcal{T}(\mu, p, q)$ can be viewed as an extreme case of the three-component Gaussian mixture $x_i \sim \mathcal{M}(\mu, \sigma^2, p, q)$ with $\sigma \to 0$. Thanks to the good generalizability of Gaussian mixture models, as will be seen in our experiments, the two properties analyzed above apply to a variety of real-world data.

### C.2.2 VALIDATING EQUATION 12 BY STATISTICALLY ESTIMATING $\mathbb{E}|\mathbf{r}^\top\mathbf{x}|/(\mu\sqrt{n/m})$ WITH SYNTHETIC DATA

Similarly as in Section C.1.2, we here verify the accuracy of Theorem 2, including the expression equation 9 of $\mathbb{E}|\mathbf{r}^\top\mathbf{x}|$ and its convergence equation 12 by performing statistical simulations on $\mathbf{x}$ and $\mathbf{r}$. The simulation results should agree with the numerical computation results P5 and P6, if the theorem is correct. In the simulation, we estimate the value of $\mathbb{E}|\mathbf{r}^\top\mathbf{x}|/\sqrt{n/m}$ by drawing $10^6$ pairs of $\mathbf{x}$ and $\mathbf{r}$ from their respective distributions and then computing the average of $|\mathbf{r}^\top\mathbf{x}|_1/\sqrt{n/m}$ as the estimate. The parameters of the distributions of $\mathbf{x}$ and $\mathbf{r}$ are set as follows: $m = 1$, $n = 10000$, $\mu = 1$, $\sigma = 1/3$ and $p = 1/2$ or $2/3$. The data dimension $n = 10000$ allows $k$ to vary between 1 and 10000. The average value of $|\mathbf{r}^\top\mathbf{x}|/\sqrt{n/m}$ at each $k$ is presented in Figures 6(c) and (d), with $p = 1/2$ and $2/3$, respectively. Comparing the numerical computation results and the simulation results shown in Figure 6, specifically contrasting (a) vs. (c) and (b) vs. (d), it can be seen that two kinds of results are roughly consistent with each other. The consistency validates Theorem 2, as well as the numerical computation results P5 and P6.

### C.2.3 VALIDATING EQUIVALENCE RELATIONS BETWEEN $\mathrm{MAD}(z)$ AND $\mathbb{E}|r^\top x|$

In Section 2.3, our statistical analysis demonstrates that the trend of $\mathrm{MAD}(\mathbf{z})$ against varying matrix sparsity $k$ can be estimated with $\mathbb{E}|\mathbf{r}^\top\mathbf{x}|$. By comparing the simulation values of $\mathrm{MAD}(\mathbf{z})$ (Figures 7 (a) and (b)) with those of $\mathbb{E}|\mathbf{r}^\top\mathbf{x}|$ (Figures 6 (c) and (d)), it is seen that they indeed present similar trends. This validates the equivalence relation between them.

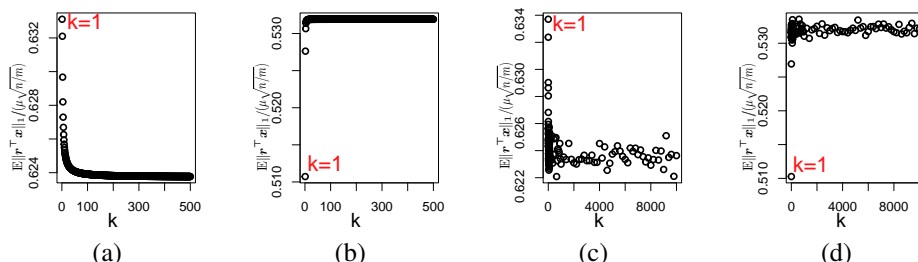

(a)  (b)  (c)  (d)

Figure 6: The value of $\mathbb{E}|\boldsymbol{r}^\top\boldsymbol{x}|/\sqrt{n/m}$ calculated by equation 9 with $p = 1/2$ (a) and $p = 2/3$ (b), and estimated by statistical simulation with $p = 1/2$ (c) and $p = 2/3$ (d), provided $x_i \sim \mathcal{M}(p, q, \mu, \sigma^2)$, $\mu = 1$ and $\sigma = 1/3$.

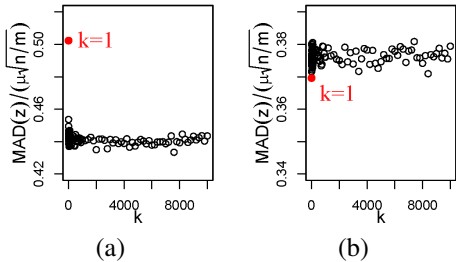

(a)  (b)

Figure 7: The value of $\mathrm{MAD}(\boldsymbol{z})$ against varying $k$ is estimated in (a) and (b), respectively, with the synthetic data generated using the parameters $p = 1/2$ and $p = 2/3$ as defined in Figures 6 (c) and (d) for $x_i \sim \mathcal{M}(p, q, \mu, \sigma^2)$, $\mu = 1$ and $\sigma = 1/3$.

# D  EXPERIMENTS

## D.1  SETTING

### D.1.1  DATA

For the sake of generality, we evaluate our classification performance across datasets with diverse attributes and scales, including the YaleB image dataset (Georghiades et al., 2001; Lee et al., 2005), the Newsgroups text dataset (Joachims, 1997), the AMLALL gene dataset (Golub et al., 1999), the MNIST binary image dataset (Deng, 2012), the CIFAR100 image dataset (Krizhevsky & Hinton, 2009), as well as the large-scale ImageNet1000 image dataset (Krizhevsky et al., 2012). While most datasets can be approximately modeled using Gaussian mixtures, the MNIST dataset follows two-point distributions. The data settings are introduced as follows. YaleB contains $168 \times 192$-sized face images of 38 persons, with about 64 faces per person. For easier simulation, we reduce the image size to $40 \times 30$. Newsgroups consists of 20 categories of text data, with 500 samples per category. Each sample is represented with a 3060-dimensional bag-of-words feature vector. AMLALL contains 25 samples taken from patients suffering from acute myeloid leukemia (AML) and 47 samples from patients suffering from acute lymphoblastic leukemia (ALL), with each sample expressed with a 7129-dimension gene vector. MNIST involves 10 classes of $28 \times 28$-sized handwritten digit images in MNIST, with 6000 samples per class and with each image pixel 0-1 binarized. CIFAR100 contains 100 categories of $32 \times 32$-sized color images, with 600 samples for each category. Among them, 5/6 of samples are used as the training set. ImageNet1000 contains 1,000 object categories, with approximately 1,000 images for each category. In total, there are about 1.2 million training images, 50,000 validation images, and 100,000 unlabeled test images. For ImageNet1000 and CIFAR100, we extract their features using the Vision-Tansformer model ViT-B/32 (Dosovitskiy et al., 2020).

### D.1.2 IMPLEMENTATION

The random projection based classification is implemented by first multiplying original data with $k$-sparse random matrices and then classifying the resulting projections with a classifier. To clearly reflect the separability of projected data, we adopt the naive $K$-nearest neighbor ($K$NN) classifier (with $K = 5$) (Cover & Hart, 1967), which relies solely on pairwise data similarities without additional discrimination enhancement. For comprehensive validation of our theoretical results, we design two experimental settings: 1) binary classification on classical datasets (Figure 1), to verify basic performance; and 2) multiclass classification on large-scale datasets (Figure 2), specifically CIFAR100 and ImageNet1000, to test scalability under complex scenarios. Similarly to $K$NN, the desired performance trends can also be obtained with other more sophisticated classifiers, like SVMs (Cortes & Vapnik, 1995), as detailed below. We also include Gaussian random projection as a baseline, given its prevalent use in the random projection research.

In the binary classification, we enumerate all possible class pairs in each dataset. For each class of data, we have one half of samples randomly selected for training and the rest for testing. To suppress the instability of random matrices and obtain relatively stable classification performance, as in (Bingham & Mannila, 2001), we repeat the random projection-based classification 5 times for each sample and make the final classification decision by voting. Usually, each classification process takes less than 0.0001 seconds on a computer equipped with an Intel Core i9-10980XE CPU and 256G of RAM. The multiclass classification adopts the testing and training sets defaulted in CIFAR100 and ImageNet1000.

### D.2 SVM CLASSIFICATION RESULTS

In Figure 8, we test the SVM (with linear kernel) classification accuracy for sparse matrices with varying matrix sparsity $k$ and projection ratio $m/n$ on four different types of datasets. It is seen that the performance changing trends of SVM against the varying matrix sparsity $k$ are similar to the $K$NN performance as illustrated in the body of the paper, thus consistent with our theoretical analysis.

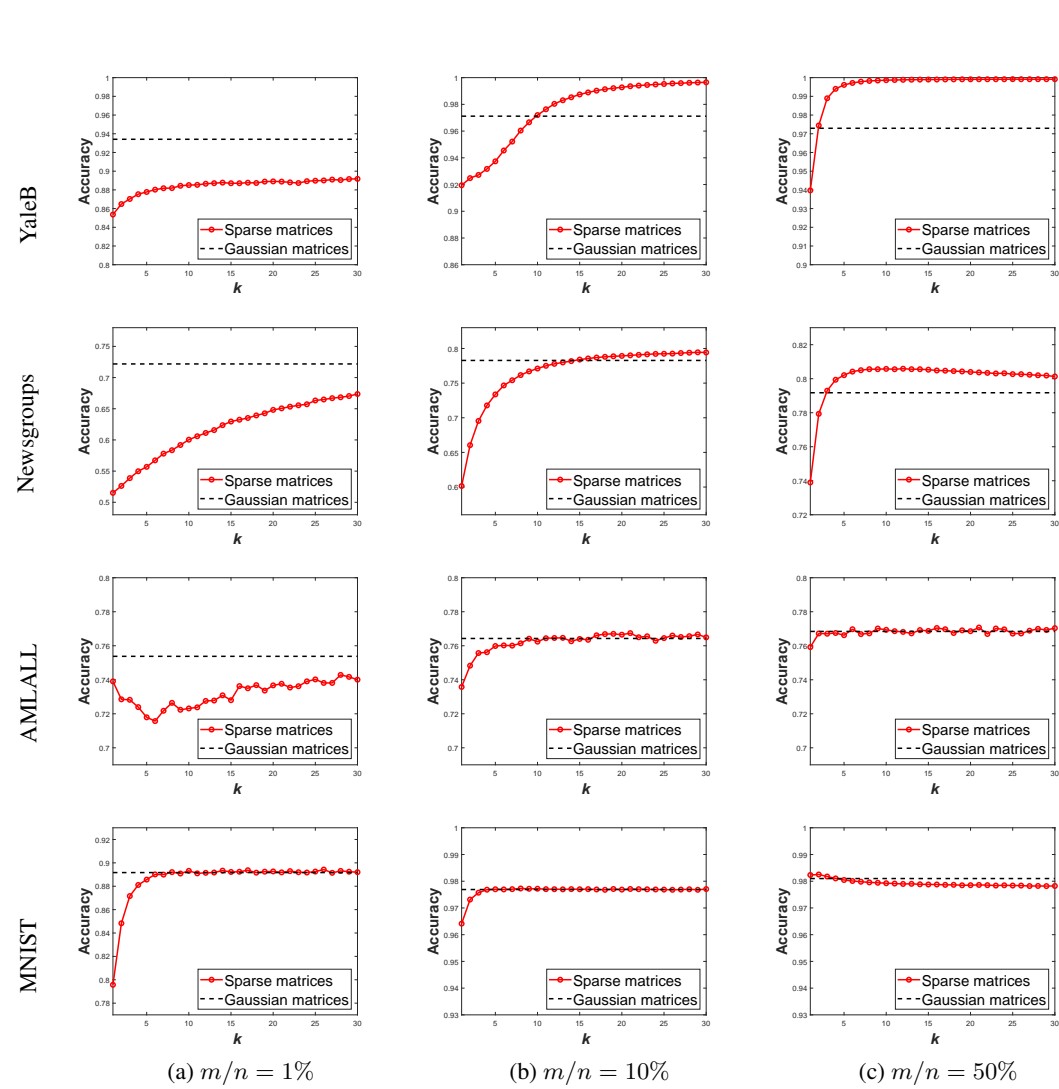

(a) $m/n = 1\%$     (b) $m/n = 10\%$     (c) $m/n = 50\%$

Figure 8: SVM classification accuracy of sparse matrix-based random projections with varying matrix sparsity $k \in [1, 30]$ and three different projection ratios $m/n = 1\%$, $10\%$ and $50\%$. The data include the image data (YaleB, DCT features), text data (Newsgroups), gene expression data (AMLALL) and binary data (MNIST, binarized pixels). The performance of Gaussian matrix-based random projections is provided as a baseline.

