# OpenReview forum: "The Sparse Matrix-Based Random Projection: Exploring Optimal Sparsity for Classification"
_ICLR.cc/2026/Conference — ICLR 2026 Conference Withdrawn Submission_

### Official Review · Reviewer_Cdb8 · 2025-10-29

**Soundness:** 2
**Presentation:** 3
**Contribution:** 1
**Rating:** 2
**Confidence:** 5

**Summary:**

This paper studies sparse $\{0,\pm1\}$ random projection matrices and investigates the optimal sparsity level for classification tasks. By analyzing the mean absolute deviation (MAD) of projected data under two representative distributions (two-point and Gaussian mixtures), the authors show that extremely sparse matrices---with only one or a few nonzero entries per row---can achieve comparable or even superior classification performance to denser matrices when $m \ge O(\sqrt{n})$. The results imply significant computational savings without loss of accuracy.

**Strengths:**

S1 (Quality): Theoretical derivations are detailed, with convergence bounds and conditions for optimal sparsity explicitly characterized. Empirical results cover diverse data modalities and scales.

S2 (Clarity): The motivation, assumptions, and implications are clearly stated, and the paper is generally well written.

S3 (Efficiency): The results imply potentially large computational savings for random-projection-based pipelines, and offer insights relevant to large models and quantized deep networks.

**Weaknesses:**

W1 (Limited Novelty): The conceptual contribution appears incremental. Extremely sparse random projections and ternary/binary structures have been extensively explored in the context of model compression and quantized neural networks, raising concerns about whether the paper provides sufficiently new theoretical insight beyond existing literature.

W2 (Strong Distributional Assumptions): The analysis relies on two idealized data models—Gaussian mixture and two-point distributions. It remains unclear how well the theoretical conclusions hold under more realistic, heavy-tailed, multimodal, or correlated distributions commonly observed in large-scale datasets.

W3 (Weak Link Between MAD and Accuracy): MAD is used as a surrogate for classification performance, but the paper does not rigorously justify why maximizing MAD directly correlates with improved accuracy. The connection is empirical and intuitive, lacking a formal link to decision boundaries, margin analysis, or downstream classifier behavior.

W4 (Limited Baselines): Experiments only compare against Gaussian random projection. Without comparisons to other state-of-the-art dimensionality reduction or projection techniques, it is difficult to evaluate the practical advantage and competitiveness of the proposed approach.

**Questions:**

Q1 (Novelty Clarification):
Prior work has extensively investigated ternary/binary projections and extreme sparsity in the context of compression and quantized neural networks. Could the authors more clearly distinguish what new theoretical insight this paper provides beyond existing analyses?

Q2 (Distributional Robustness):
The theoretical analysis assumes Gaussian mixture and two-point distributions. How robust are the results if data deviates from these assumptions (e.g., heavy tails, multimodal density, correlated features)? Can the authors provide empirical evidence, theoretical arguments, or discussion on how far the conclusions generalize under real-world distributional shifts?

Q3 (Justification of MAD as a Surrogate):
MAD is used as a proxy for classification accuracy, but the paper currently offers only intuitive justification. Can the authors provide a more rigorous argument or reference connecting MAD to downstream accuracy—such as its relationship to class separability, classifier margins, or error bounds? Under which conditions might maximizing MAD not translate into improved accuracy?

Q4 (Baseline Coverage):
Experiments only compare against Gaussian random projection. To better evaluate practical competitiveness, could the authors include additional baselines?

---

> ### Author Response · Authors · 2025-11-12
> **Response to Reviewer Cdb8**
>
> **Response to Q1 (Novelty Clarification):**  As we have emphasized in the Abstract and Introduction, we are the first to estimate optimal sparsity of sparse ternary matrices-based random projection,  and our study reveals that extremely sparse matrices (with only one or a few tens of nonzero entries per row) can match or outperform the performance of other denser matrices.  This finding allows us to significantly reduce the  complexity of random projection. **So the contribution is significant for the efficient applications of random projection.**
>
> **Response to Q2 (Distributional Robustness):**  In the experiments, we have tested different types of real-world data features, like DCT, DWT, Deep Convolution features, and Inverse Document Frequency (of text)， which usually have sparse (heavy-tailed)  distributions, rather than exactly Gaussian distributions. Despite the distribution differences, as discussed in our experiments, **these experimental results are highly consistent with our theoretical predictions**.  This suggests that our theoretical results have strong generalizability.
>
> **Response to Q3 (Justification of MAD as a Surrogate):** The argument that "larger MAD values (dispersion/variation) probably lead to superior features for classification " has been widely validated by the early research on PCA. See the references given in our Introduction for details.  In fact,  this property is also the base of PCA to be used for dimensionality reduction.
>
> Regarding the precise/theoretical relationship between  dispersion/variation and classification, as far as we know,  it remains a challenging and unresolved problem, despite the long history of PCA development.
>
> **Response to Q4 (Baseline Coverage):**  Note that our focus is to estimate the optimal sparsity of ternary matrices for random projection-based classification.  This issue has been thoroughly examined  in our experiments.  The comparison with other dimension reduction methods (beyond random projection) is out of our research scope.

---

### Official Review · Reviewer_W43D · 2025-10-31

**Soundness:** 3
**Presentation:** 3
**Contribution:** 3
**Rating:** 6
**Confidence:** 4

**Summary:**

The paper proposes a new random projection method, where the projection matrix is constructed by $\{0, \pm 1\}$. The authors establishes a connection between matrix sparsity and classification performance via MAD analysis, providing a new perspective beyond traditional distance preservation. It seems that the analysis in this work relies heavily on specific data distributions. Therefore, I recommend weak accept.

**Strengths:**

The authors conduct numerical experiments using six datasets (images, text, genes, binary) to support the theoretical claim.

**Weaknesses:**

The limitation of the assumption for the original data: The analysis relies heavily on specific data distributions (e.g. Gaussian mixture in the manuscript)

**Questions:**

How to generalize your framework to more general original data distributions?

---

> ### Author Response · Authors · 2025-11-12
> **Response to Reviewer W43D**
>
> **Response (the generalization to real-world data):**  In the experiments, we have tested different types of real-world data features, like the DCT, DWT, Deep Convolution, and binarized features (of images),  the Inverse Document Frequency (of text)，and the gene data,  which usually have sparse (heavy-tailed)  distributions, rather than exactly Gaussian distributions. Despite the distribution differences, as discussed in our experiments, **these experimental results are highly consistent with our theoretical predictions**.  This suggests that our theoretical results have strong generalizability.

---

### Official Review · Reviewer_9s5R · 2025-11-03

**Soundness:** 1
**Presentation:** 2
**Contribution:** 1
**Rating:** 2
**Confidence:** 4

**Summary:**

The manuscript suggests that sketching with sparse matrices tends to preserve classification accuracy in practice. Motivated by this observation, the authors study the maximum absolute deviation (MAD) of the projected data and argue that this framework explains the effectiveness of sparse sketching compared to dense matrices, such as Gaussian ones.

**Strengths:**

The paper reads well.

**Weaknesses:**

I think certain parts of the paper should be explained better, and some statements appear to be incorrect:

- In lines 58–61, the authors suggest that sketching with sparse matrices preserves accuracy well and often performs comparably to or better than dense sketching. However, no citation is provided. A citation should be provided, as this is the motivation of the paper.
- I could not follow why MAD is a good metric for understanding classification performance. The authors suggest that MAD is a more robust metric for quantifying dispersion; however, neither the text nor the examples involve heavy-tailed models or outliers. As written, the claim that MAD is a better alternative is not convincing.
- There are some incorrect statements in the paper that should be addressed:
  - In line 175, it is argued that when the original data $h$ follow the Gaussian mixture distribution described above, the projected data $z$ remain Gaussian. This appears to be incorrect.
  - In line 179, it is argued that this relationship also holds approximately for original data $h$ drawn from other distributions, since by the Central Limit Theorem, the projected data $z \in \mathbb{R}^m$ can be approximated by a Gaussian distribution. This claim is too broad, since it depends heavily on the dimension and on the tail properties of the data in high-dimensional settings.
  - In line 364, the authors suggest that MNIST follows a two-point distribution. On what basis? Please clarify.

**Questions:**

See above

---

> ### Author Response · Authors · 2025-11-12
> **Response to Reviewer 9s5R**
>
> **Comment 1:** In lines 58–61, the authors suggest that sketching with sparse matrices preserves accuracy well and often performs comparably to or better than dense sketching. However, no citation is provided. A citation should be provided, as this is the motivation of the paper.
>
> **Response 1:**  The relevant experiments can be found in the Table 2 of the paper "Quantization: Is It Possible to Improve Classification?", IEEE DCC 2023.
>
> **Comment 2:** I could not follow why MAD is a good metric for understanding classification performance. The authors suggest that MAD is a more robust metric for quantifying dispersion; however, neither the text nor the examples involve heavy-tailed models or outliers. As written, the claim that MAD is a better alternative is not convincing.
>
> **Response 2:**  In general, the dispersion is quantified with two metrics: the $\ell_2$ norm-based variance, as employed in PCA, and the $\ell_1$ norm-based the mean absolute deviation (MAD) (Yager & Alajlan, 2014). Compared to the variance, MAD is recognized for its greater robustness to noise and outliers, enabling a more accurate estimation for the dispersion of
> real-world data (Deng et al., 2007; McCoy & Tropp, 2011; Meng et al., 2012; Hubert et al., 2016).  Overall, the advantage of MAD has been early studied and proved in  these references.
>
> **Comment 3：**  In line 175, it is argued that when the original data $h$ follow the Gaussian mixture distribution described above, the projected data $z$ remain Gaussian. This appears to be incorrect.
>
> **Response 3:**  Thank you for pointing out this problem with Property 1. Now we have found that Property 1 is not necessary for us to analyze MAD.  Removing Property 1 would require substantial modifications to the entire manuscript. In light of this, we wish to withdraw the current submission and publish it in the future.
>
> **Comment 4:**  In line 179, it is argued that this relationship also holds approximately for original data  $h$ drawn from other distributions, since by the Central Limit Theorem, the projected data  $z$ can be approximated by a Gaussian distribution. This claim is too broad, since it depends heavily on the dimension and on the tail properties of the data in high-dimensional settings.
>
> **Response 4:**  Yes. The claim about Gaussian-distributed $z$ tends to hold for high-dimensional data $h$. As we know, there is always a gap between theory and practice.
>
> **Comment 5:** In line 364, the authors suggest that MNIST follows a two-point distribution. On what basis? Please clarify.
>
> **Response 5:** The images in MNIST have pixels taking binary values 0 and 1, corresponding to the dark background and the while characters, respectively.

---

### Official Review · Reviewer_H7au · 2025-11-04

**Soundness:** 2
**Presentation:** 2
**Contribution:** 3
**Rating:** 4
**Confidence:** 3

**Summary:**

This paper investigates the optimal sparsity level for $\{0, \pm 1}$-matrix random projections in classification tasks by analyzing the dispersion of projected data points under two representative data distributions. The main finding is that extremely sparse matrices with only one or a few nonzero entries per row can match or exceed the classification performance of much denser alternatives, offering significant computational savings. This result is validated experimentally across diverse real-world datasets including images, text, gene expression data, and binary data.

**Strengths:**

- The paper tackles why extremely sparse random projections can still perform well in classification despite violating traditional distance-preservation guarantees, which is an interesting problem to tackle.
- The proposal to use Mean Absolute Deviation (MAD) as a proxy for classification performance seems to be a novel departure from standard $l_2$ or variance-based analyses. It potentially provides a new lens through which to analyze the problem.
- The theoretical findings are tested against a wide and diverse range of datasets, including images (YaleB, CIFAR100, ImageNet1000), text , and gene data.

**Weaknesses:**

1. **Connection between Disperson and Classification:** A core concern I have about the proposed view is the connection between dispersion and classification. The authors mention in the abstract that, *"Statistically, a higher degree of dispersion is expected to improve classification performance by capturing more intrinsic variations in the original data."*  However, I could not find anywhere in the paper where the author theoretically examines this. Since, this is a theory paper proposing this central hypothesis of MAD being a proxy for classification, I think this should be theoretically explored.
    - I am also not convinced that PCA is a good example when supporting this view, as the goal of PCA is reconstruction and not class separability.
   - If dispersion is the metric, it means everything is dispersed by the projection. I think inter/between-class variance relative to intra/within-class variance would be the proper metric for classification? How does MAD capture these nuances and how do the authors study this point?

2. **Claim about $z$ being Gaussian:**  The paper invokes the closure of Gaussian distributions under linear transformations to claim that the projected variable $z = Rh$ “remains Gaussian.” However, in section 2.2.2 $h$ is modeled as a Gaussian mixture (not Gaussian). Hence, $z$ is, in general, a *mixture* of Gaussians rather than a single Gaussian. This misapplication of the closure property undermines the validity of later derivations that rely on i.i.d. Gaussian assumptions (e.g., the MAD identity). Can the authors clarify?

3. In terms of proof presentation, the central point of the proof could be summarized much more neatly in the main paper and more of the technical details could be stowed away in the appendix.

**Questions:**

Please, see the weakness section.

---

> ### Author Response · Authors · 2025-11-14
> **Response to Reviewer H7au**
>
> **Comment 1:**  Connection between Disperson and Classification: A core concern I have about the proposed view is the connection between dispersion and classification. The authors mention in the abstract that, "Statistically, a higher degree of dispersion is expected to improve classification performance by capturing more intrinsic variations in the original data." However, I could not find anywhere in the paper where the author theoretically examines this. Since, this is a theory paper proposing this central hypothesis of MAD being a proxy for classification, I think this should be theoretically explored.
>
> **Response 1:** In the research of principal component analysis (PCA) (Jolliffe & Cadima, 2016), it is argued that the projections with higher dispersions probably yield superior features for classification, as they capture more intrinsic variations in the original data. This characteristic has been thoroughly validated through the wide applications of PCA for dimension reduction. In general, the dispersion
> is quantified with two metrics: the  $\ell_2$ norm-based variance, as employed in PCA, and the $\ell_1$ norm-based the
> mean absolute deviation (MAD) (Yager & Alajlan, 2014). Compared to the variance, MAD is recognized
> for its greater robustness to noise and outliers, enabling a more accurate estimation for the dispersion of
> real-world data (Deng et al., 2007; McCoy & Tropp, 2011; Meng et al., 2012; Hubert et al., 2016).
>
> **Comment 2:** I am also not convinced that PCA is a good example when supporting this view, as the goal of PCA is reconstruction and not class separability.
>
> **Response 2:**  In machine learning, PCA is mainly used as an unsupervised/linear dimensionality reduction technique, similar to random projection.
>
> **Comment 3:** If dispersion is the metric, it means everything is dispersed by the projection. I think inter/between-class variance relative to intra/within-class variance would be the proper metric for classification? How does MAD capture these nuances and how do the authors study this point?
>
> **Response 3:** Note that we cannot consider the inter-/intra-classes scattering ratios for random projection (and PCA), because they are *unsupervised* dimension reduction techniques. For random projection and PCA, the study is focused on the variation/dispersion in the projected data. As for the reason, we have briefly discussed in Response 1. For details,  please see  relevant references.
>
> **Comment 4:** Claim about $z$ being Gaussian: The paper invokes the closure of Gaussian distributions under linear transformations to claim that the projected variable  $z=Rh$ “remains Gaussian.” However, in section 2.2.2 $h$
>  is modeled as a Gaussian mixture (not Gaussian). Hence, $z$ is, in general, a mixture of Gaussians rather than a single Gaussian. This misapplication of the closure property undermines the validity of later derivations that rely on i.i.d. Gaussian assumptions (e.g., the MAD identity). Can the authors clarify?
>
> **Response 4:**  **Thank you for pointing out this problem with Property 1. Now we have found that Property 1 is not necessary for us to analyze MAD.  Removing Property 1 would require substantial modifications to the entire manuscript. In light of this, we wish to withdraw the current submission and publish it in the future.**

---

### Note · Authors · 2025-11-14

**Comment:**

Dear Chairs and Reviewers,

We would like to thank you for taking  the time to review our paper. Unfortunately, yet fortunately, two reviewers have identified an error in our Property 1. While we can devise a solution to resolve this problem, it would require substantial modifications to the entire manuscript. In light of this, we wish to withdraw the current submission and publish it in the future.

Best regards,

Authors

**Withdrawal Confirmation:**

I have read and agree with the venue's withdrawal policy on behalf of myself and my co-authors.